# Research on Uncertainty of Landslide Susceptibility Prediction—Bibliometrics and Knowledge Graph Analysis

**Zhengli Yang** [1,2], **Chao Liu** [1,2], **Ruihua Nie** [1,2], **Wanchang Zhang** [3], **Leili Zhang** [1,2], **Zhijie Zhang** [4], **Weile Li** [5], **Gang Liu** [6], **Xiaoai Dai** [6], **Donghui Zhang** [7], **Min Zhang** [8], **Shuangxi Miao** [9], **Xiao Fu** [10], **Zhiming Ren** [11] and **Heng Lu** [1,2,*]

1   State Key Laboratory of Hydraulics and Mountain River Engineering, Sichuan University, Chengdu 610065, China
2   College of Hydraulic and Hydroelectric Engineering, Sichuan University, Chengdu 610065, China
3   Key Laboratory of Digital Earth Science, Aerospace Information Research Institute, Chinese Academy of Sciences, Beijing 100094, China
4   School of Geography, Development & Environment, University of Arizona, Tucson, AZ 85721, USA
5   State Key Laboratory of Geohazard Prevention and Geoenvironment Protection, Chengdu University of Technology, Chengdu 610059, China
6   College of Earth Science, Chengdu University of Technology, Chengdu 610059, China
7   Aerospace Information Research Institute, Chinese Academy of Sciences, Beijing 100094, China
8   School of Environmental and Geographical Sciences, Shanghai Normal University, Shanghai 200234, China
9   College of Land Science and Technology, China Agricultural University, Beijing 100091, China
10   Faculty of Geosciences and Environmental Engineering, Southwest Jiaotong University, Chengdu 611756, China
11   Sichuan Zhide Geotechnical Survey Co., Ltd., Chengdu 610042, China
*   Correspondence: luheng@scu.edu.cn

**Abstract:** Landslide prediction is one of the complicated topics recognized by the global scientific community. The research on landslide susceptibility prediction is vitally important to mitigate and prevent landslide disasters. The instability and complexity of the landslide system can cause uncertainty in the prediction process and results. Although there are many types of models for landslide susceptibility prediction, they still do not have a unified theoretical basis or accuracy test standard. In the past, models were mainly subjectively selected and determined by researchers, but the selection of models based on subjective experience often led to more significant uncertainty in the prediction process and results. To improve the universality of the model and the reliability of the prediction accuracy, it is urgent to systematically summarize and analyze the performance of different models to reduce the impact of uncertain factors on the prediction results. For this purpose, this paper made extensive use of document analysis and data mining tools for the bibliometric and knowledge mapping analysis of 600 documents collected by two data platforms, Web of Science and Scopus, in the past 40 years. This study focused on the uncertainty analysis of four key research subfields (namely disaster-causing factors, prediction units, model space data sets, and prediction models), systematically summarized the difficulties and hotspots in the development of various landslide prediction models, discussed the main problems encountered in these four subfields, and put forward some suggestions to provide references for further improving the prediction accuracy of landslide disaster susceptibility.

**Keywords:** landslide; susceptibility prediction; uncertainty analysis; VOSviewer; Ctiespace; bibliometric analysis; knowledge graph

## 1. Introduction

A landslide is a phenomenon in which the rock and soil mass on a slope slides down as a whole or dispersedly along a particular weak surface (belt) under the action of gravity due to the influence of an earthquake [1], rainfall [2], hurricane [3], snow melt [4],

human activities [5], tsunami, and other factors [6]. As one of the main types of geological disasters [7], a landslide is the second largest natural disaster after earthquakes [8] and can pose a significant threat to human life and property, infrastructure, and the natural environment because of its substantial destructive capacity [9]. Landslides cause billions of dollars of property losses and thousands of deaths yearly [10]. Statistics show that the number of deaths caused by landslides accounts for at least 17% of that caused by natural disasters worldwide [11]. Of these, earthquake-induced landslides have killed more people than any other type of landslide [1]. For example, the Wenchuan Earthquake (a magnitude-8.0 earthquake) in May 2008 triggered 197,481 landslides in many regions [12]. Its total affected area was about 1160 km$^2$, and the number of deaths was more than 25,000 [12,13], accounting for a quarter of the total casualties in this accident [14]. In addition, according to Froude, M, J. et al. [15], the spatiotemporal analysis of the global non-seismic fatal landslide data set from 2004 to 2016 showed that there were 55,997 deaths in 4862 different landslide events. Typical non-seismic landslide hazard events include events such as, in October 1998, hurricane Mitch caused catastrophic landslides in the Caribbean and Central America, leaving 6600 people dead and 8052 others injured [16]. In February 2006 [17], after several days of heavy rainfall, a large-scale landslide almost buried the Jinsuogong Village on Leyte Island in the Philippines, causing at least 1800 people to die, 3264 people to move far away from their homes, and 18,862 to be affected in some way.

The question of how to reduce the loss caused by landslide disasters through prediction has become a hot issue concern of researchers [18–21]. As the first step of landslide risk analysis, the prediction of landslide susceptibility is to make a qualitative or quantitative analysis of landslide disasters in a particular area to find out the combination of factors most conducive to the occurrence of landslides [22]. Then, these combinations of factors are used to predict the possibility of the occurrence of landslide disasters in areas of the same type to determine the scope in which the landslides may occur. The successful landslide prediction can significantly reduce the disaster-affected degree, can avoid the occurrence of landslide disasters [23], and can provide time and conditions for human beings to take early action before the event of landslide disasters to hinder the occurrence of risks and prevent such danger from becoming a disaster to human beings [24]. Some typical cases of successful prediction are listed below. In 1985, a large-scale landslide occurred in Xintan town of the Three Gorges Reservoir Region, but due to the early monitoring and forecast of the landslide and the well-organized evacuation, none of the 1,371 residents were killed or injured [25]. As researchers successfully predicted the Heifangtai landslide in advance [26,27], and the local government took active disaster prevention measures in time according to the early warning results, no one was killed or injured by the landslide in this area. Since Japan developed an early warning system for landslides that can protect people from landslide damage and property loss through landslide prediction in 1984, the damage caused by landslides has been significantly less than that in the past [28].

Although the uncertainty of the catastrophic process of the landslide has become the scientific community's consensus, no mature theories or methods have been formed to predict landslide susceptibility, and many studies are still in the exploratory stage. Therefore, the evaluation of uncertainty is of vital importance in the prediction and analysis of landslide susceptibility. In the past 40 years, scholars have established landslide disaster prediction models for landslide susceptibility prediction by different calculation methods, providing important references for landslide prediction and analysis. Regarding the summarization of forecast and analysis of landslide susceptibility, some review papers on landslide prediction from different disciplines have appeared one after another [29–32]. Through the study of the review papers on landslide prediction, it is not difficult to find that the existing research mainly focuses on the evaluation of landslide susceptibility models [33–35]. There is less literature reviews on the uncertainty analysis of landslide susceptibility prediction [36], and only Reichenbach, P. et al. have conducted research on the document analysis of the uncertainty of landslide susceptibility prediction through bibliometric analysis and knowledge mapping analysis [37]. This research mainly summarizes

the statistical methods of landslide sensitivity modeling and related topographic zoning but rarely involves uncertainty analysis. In addition, there is almost no comprehensive research on the spatial and temporal variation law of the uncertainty analysis of landslide susceptibility prediction. Based on the summary of previous research, and considering that the Web of Science (WOS) data platform began to track this field in 1992 and the Scopus data platform published papers in this field from 1982, this paper retrieved the two data platforms of WOS and Scopus to provide more comprehensive information. Therefore, this paper created a bibliometric analysis of 600 selected documents, quantified document performance, and analyzed highly cited documents by the data analysis functions of the WOS and Scopus platforms. Meanwhile, it comprehensively used visual software of VOSviewer and CiteSpace to draw the knowledge mapping and interpreted it in combination with professional knowledge to clarify the knowledge structure and development trend in this field, mine and analyze research hotspots, and predict the future development direction. Moreover, it systematically summarized the uncertainty of landslide susceptibility prediction from four subfields (namely disaster-causing factors, prediction units, model space data sets, and prediction models). It put forward some suggestions to provide some references for further research.

## 2. Data Sources and Analysis Methods

### 2.1. Retrieval Strategy and Data Collation

WOS is a comprehensive academic information resource that ranks first globally and covers most disciplines, while Scopus is the world's largest abstract and citation database. This paper retrieved data from WOS and Scopus platforms for analysis on 14 June 2022. As shown in Figure 1, the retrieval strategy was TS = "landslide" AND "predict*" AND "uncertainty", of which "predict*" contains two subject words: "predict" and "prediction". First, in the WOS document data platform, the data source was set as "Search in: Web of Science Core Collection". Next, the citation index was selected as "Editions: All", the document field selected as "Topic", and the retrieval time set as "Publication Date: All years (1900–2022)". A total of 459 records were retrieved from WOS. Then, the Scopus document data platform was logged into again, the retrieval range was set as "paper title, abstract, keyword", and the retrieval time was selected from "all years" to "till now", and 453 records were retrieved. A total of 912 papers were recovered from the two document data platforms. In order to ensure the accuracy and relevance of documents and to facilitate the bibliometric and knowledge mapping analysis, the original data needed to be downloaded in different formats and provided with data preparation work such as format unification, collation, de-duplication, and item-by-item screening. The specific process is shown in Figure 1.

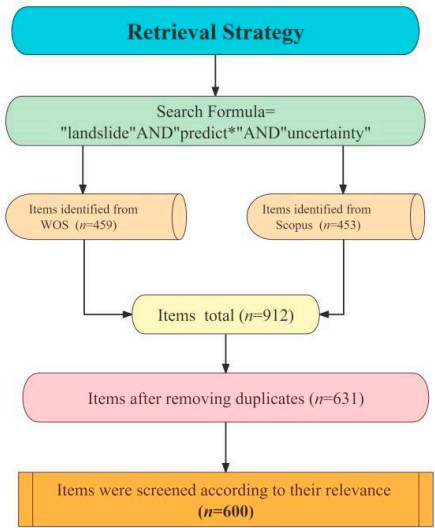

**Figure 1.** Flowchart of the process of searching and selecting the studies.

1. Unified format: The text data format (*.ris, *.txt) and table data format (*.xls, *.csv) were adopted for download, respectively. Endnote mainly uses the data in *.ris format for document browsing and reference management, and it is also convenient for VOSviewer software to make bibliometric analyses and visual analyses. The files in *.txt format were named download_w459.txt (from WOS) and download_s453.txt (from Scopus). The synonyms were merged, and wrong words were corrected for knowledge mapping analysis by Citespace. The two data formats of *.xls and *.csv were unified as *.xls to prepare for the next step of data collation, de-duplication, and screening.

2. Collation and de-duplication: In these two platforms, all original data were downloaded to the Excel spreadsheet in the way of the full record, were de-duplicated according to the title and DOI after being combined into a file, were extracted according to the year, author, title, publication type, journal source, number of citations, keywords, and other information, and then sorted by the publication type as the first column.

3. Item-by-item screening: Manual screening was carried out item by item according to the publication type of the documents, and only the documents with peer review and editor's supervision were retained. The types of these documents include articles, reviews, letters, and conference papers. The editorial material, book chapter, and other documents that were weakly correlated with this paper were removed. After identification and iteration, some wrong records were deleted to finally obtain 600 valid documents. The typical document characteristics are shown in Table 1.

**Table 1.** Characteristics of the included studies.

| Publication Type | Year | Authors | Title | Publication/ Source Titles | Cited Reference Count | Keywords |
|---|---|---|---|---|---|---|
| Article | 2017 | Park, H.J. et al. [38] | Physically based susceptibility assessment of rainfall-induced shallow landslides using a fuzzy point estimate method | Remote sensing | 141 (WOS) | Monte-carlo-simulation; differential sar interferometry; rock slope stability; li-shan landslide; modeling uncertainty; reliability-analysis; hazard assessment; satellite; risk; failure |
| Review | 1996 | Johnston, A.C. [39] | Seismic moment assessment of earthquakes in stable continental regions—III. New Madrid 1811-1812, Charleston 1886, and Lisbon 1755. | Geophysical Journal International | 352 (Scopus) | Earthquake intensity; earthquake-source mechanism; seismic moment |
| Letter | 2022 | Ho, J.Y. et al. [40] | Using ensemble quantitative precipitation forecast for rainfall-induced shallow landslide predictions | Geoscience letters | 23 (WOS) | Physically-based model |
| Conference Paper | 2021 | Nuryanto, D.E. et al. [41] | Prediction of soil moisture and rainfall induced landslides; a comparison of several PBL parameters in the WRF model | IOP Conference Series: Earth and Environmental Science | 1 (Scopus) | landslide; rainfall; soil moisture; WRF model |

### 2.2. Analysis Method

This paper mainly carried out research using two methods: bibliometric analysis and knowledge mapping analysis. The concept of bibliometric analysis can be traced back to 1969 and was proposed by Alan Pritchard [42], a famous British information scientist. It describes and evaluates published research by a quantitative method, which can help researchers to find the most influential works and objectively present the scientific structure relationship of a specific research field [43]. This paper extensively used the advantages of various bibliometric analysis tools (WOS, Scopus, VOSviewer, and SCImago Graphica) in processing data. It summarized and reviewed the research status of uncertainty analysis of landslide susceptibility by quantifying the performance of documents. VOSviewer, a tool developed by Van Eck N and Waltman L of the Centre for Science and Technology

Studies of Leiden University in the Netherlands to build a bibliometric network [44], has been widely used in all kinds of "co-occurrence" analyses. As a free chart-making software, SCImago Graphica can generate various charts only by dragging, rather than using any formula and complex data processing and modeling, which is suitable for lightweight data applications [45]. In this paper, the three aspects of statistics of documents publication time, contribution analysis, and analysis of highly cited documents were mainly analyzed by the bibliometric function of WOS and Scopus data platforms. Taking the number of regional papers published as one of the contribution analysis units, the data was imported into Scimago software for analysis after the format conversion by the collinear analysis function of the VOSviewer (version 1.6.16) software.

Knowledge mapping analysis is a method of describing knowledge resources and their carriers with visualization technology [46], and it can show the development process and structural relationship of complex knowledge [47]. To improve the accuracy and reliability of the results by knowledge mapping analysis, this paper mainly adopted two kinds of knowledge mapping analysis software, VOSviewer (version 1.6.16) and Citespace (versions 5.8.R3 and 6.1.R2, which have different effects when displaying different mappings) to solve the following three problems: (1) know the scientific research cooperation status of authors, countries and regions, and institutions; (2) identify potential key keyword nodes and mine and analyze hot research issues; (3) analyze the development process and predict the frontier trend of research. Considering that the two kinds of software of VOSviewer and Citespace can draw maps with a large amount of information and good visual effect, they can provide scientific research perspectives from different aspects. The primary motivation for using CiteSpace in this research was to simplify the search for essential papers in the knowledge domain documents so that visually significant characteristics can be searched in the optical network, and visual aids can be provided to identify the changes between adjacent nodes [48–50]. CiteSpace is a Java application developed by Professor Chen Chaomei of Drexel University in the United States for the visual analysis of the co-occurrence networks [51], which can effectively reveal the hot spots, trends, and development evolution of specific research fields [3]. Based on the above advantages, CiteSpace can be used for scientific research cooperation analysis, research hotspots analysis, and frontier trend analysis. However, as CiteSpace has the problems of overlapped keyword node names and poor visual effect when performing keyword clustering of research hotspot analysis [52], VOSviewer with a better visual effect of "collinear clustering "was selected. Moreover, the data used for keyword clustering analysis in this paper were from the two data platforms of WOS and Scopus and formed a self-built data text after collation and de-duplication. Therefore, importing a VOSviewer that can support self-built text data to extract subject words and carry out co-occurrence clustering in a single line text based on subject words is necessary. A series of knowledge mapping analyses can be achieved by comprehensively using the two kinds of software and giving full play to their respective advantages in processing data.

## 3. Results

### 3.1. Bibliometric Analysis

#### 3.1.1. Statistics of Documents Publication Time

During data collation and de-duplication, this paper retained the documents downloaded from the WOS data platform and deleted the duplicated data downloaded from the Scopus data platform but sourced from the WOS. When the statistics of record publication time is made, the original data downloaded from the WOS and Scopus and the data after de-duplication are displayed on a chart, as shown in Figure 2. Orange represents the number of documents published on the WOS platform, purple represents the number of papers published on the Scopus platform, and green represents the number of documents published after the merger and de-duplication of the two platforms. Through comparison, it was found that the results of the original documents retrieved from the two databases were different, which was mainly manifested in:

(1)   Different start times of tracking: Scopus collected the probability assessment published by Atkinson, G.M. et al. on modeling and predicting landslides due to the possibility of liquefaction and overstress caused by earthquakes from 1982 [53], while WOS collected the investigation of BUISSON, L. et al. on the application of artificial intelligence (AI) technology to environmental protection in France from 1992 [54]. This investigation first showed that landslide prevention data could be embedded into artificial intelligence systems.

(2)   Different distribution of the number of annual documents published: The years when the data on the WOS platform were more than those on the Scopus platform include 1997, 2002, 2008–2010, and 2017–2022. The years when the data on the Scopus platform were more than those on the WOS platform include 1982, 1985, 1986, 1992, 2003–2006, and 2011–2016. The two platforms have the same number of documents published in other years.

(3)   Different collection processes: The collection time of the Scopus platform is earlier than that of the WOS platform, and the number of documents published in most of the early years on the Scopus platform is more than that on the WOS platform. However, in recent years, WOS has focused more on this field and will continue to strengthen its tracking in this field in the future (seen from the trend of documents published). Therefore, the statistics of document publication time show that the research data in this field shall be downloaded from the two document platforms to make the data more comprehensive. The document sample database established based on this is more reasonable and effective.

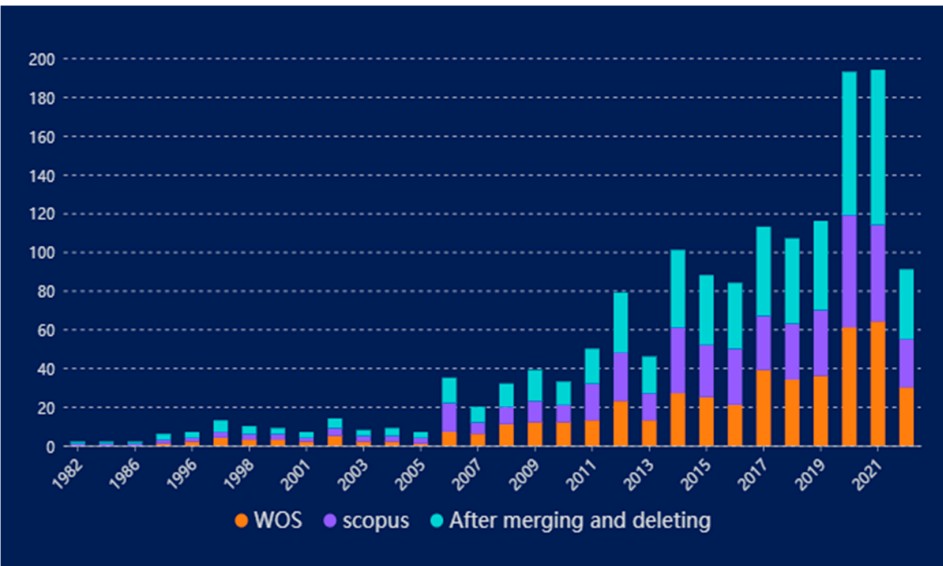

**Figure 2.** Publication time statistics.

According to the number of documents published after merging and de-duplicating in Figure 2, the research in this field over the past 40 years can be roughly divided into three stages. The first stage is the rise stage (1982–2005): the total number of documents published in this 24-year period was only 22, and the research in this field was intermittent from 1982 to 1998. Although there were only a few published documents, the most enlightening document published by Carrara, A. et al. in 1992 on attempts to evaluate landslide hazards and risks by landslide identification and mapping appeared during this period [55], which not only compared the inherent uncertainty of landslide maps drawn by different researchers or through different technologies but also proposed that the impact of uncertainty on landslide spatio-temporal prediction is an issue that requires interdisciplinary efforts. The publication time of this document coincided with the time when the WOS initially began to track the research in this field, and the number of this

document cited was as high as 156, indicating that it has high academic authority and a great influence on subsequent research. Since 2000, papers have been published yearly, and the research in this field has been continuous. People began to realize that it is difficult to predict whether landslides will occur in the future, and management decisions are often made under uncertain conditions. Therefore, landslide disaster prediction and mapping are necessary conditions for decision-making [56]. The second stage is the apparent growth stage (2006–2016): 235 documents were published during these 11 years, which is 10 times that in the past 23 years. The research theories and methods in this stage were further developed, the research perspective was gradually expanded to hot research topics such as geographic information system (GIS) [57] and susceptibility assessment [58], and analytic hierarchy process [59], multi-criteria decision analysis [60], and artificial neural network [61] began to be widely used in the field of landslide susceptibility prediction. The number of documents published in this stage had increased significantly compared with that in the rise stage of research, but the number of documents published every year was less than 40, indicating that the research in this stage is still relatively slow. The third stage is the vigorous development stage (2017–2022): the total number of published documents reached 326 in less than six years, accounting for 54.33% of the total samples. The number of papers published every year was more than 40, and the surge in the number of papers was closely related to the innovation and development of this field during this stage to a certain extent. In 2021, the number of documents published reached a peak of 83. In 2022, due to the limited retrieval time, only the number of documents published in less than half a year was counted, but it can be predicted that the number of documents published in 2022 will remain stubbornly high, indicating that the research in this field has broad prospects in the future.

### 3.1.2. Contribution Analysis

In this paper, the contribution analysis was mainly carried out by bibliometric analysis from three aspects: high productivity authors, high productivity countries or regions, and high productivity journals. In terms of the author's contribution to this field, as shown in Table 2, Guzzetti, F. et al. had the highest number of documents published [62]. In 2002, they published documents on the evaluation of the impact of landslide inventory errors on landslide disaster prediction models, showing that the error of input data is still the main bottleneck of landslide disaster prediction reliability, but statistical modeling greatly reduces the impact caused by input data errors. Another contribution of this team was to propose a framework for the reliability and prediction ability of regional landslide sensitivity and individual assessment models [63]. This achievement has been cited as high as 526 times, so they are also an author group with the highest index. The top five authors and their representative works are listed in Table 2.

To improve the visual effect of a country or regional distribution, this research first saved the merged and de-duplicated data of WOS and Scopus in a unified format of *.txt and set the word frequency threshold as 20 after the data were opened in VOSviewer to obtain the national or regional calculation results. Then, the data were saved in *.gml format and imported into Scimago to obtain the geographical visualization map of geographical document distribution after map decoration. Figure 3 shows the distribution of countries ranked at the world's top in this field. A total of 12 countries have published more than 20 documents. The top three countries are China (172, accounting for 28.67% of the total number of samples), Italy (106, accounting for 17.67% of the total number of samples), and the United States (88, accounting for 14.67% of the total number of samples), which fully shows that the three countries have obvious research advantages in this field. In addition, the other 64 countries have a small share, which is not shown in the figure. This means that many countries and regions in the world are landslide-prone areas or have published papers in this field, but it may be challenging to carry out extensive research in this field due to the limitations of their economic development level and the allocation of national scientific research funds.

**Table 2.** Prolific author (top five).

| Authors | Post Volume | Masterpiece | Representative Contribution | Index (Source, Scopus) |
|---|---|---|---|---|
| Guzzetti, F. et al. [63] | 16 | Estimating the quality of landslide susceptibility models | A landslide susceptibility model for a region in central Italy is presented, and a framework for assessing model reliability and forecasting skills is presented. | 64 |
| Tang, H. et al. [64] | 12 | A new framework for characterizing landslide deformation: a case study of the Yu-Kai highway landslide in Guizhou, China | A new framework for characterizing landslide deformation is proposed, which can establish the evolution of landslide deformation in both the geometric and temporal domains, allowing the evaluation of the sliding mechanism of landslides. | 37 |
| Gariano, S.L. et al. [65] | 12 | How much does the rainfall temporal resolution affect rainfall thresholds for landslide triggering? | The impact of the temporal resolution of rainfall measurements on a landslide-triggered rainfall threshold calculation in a region of northern Italy was assessed and discussed. | 19 |
| Peruccacci, S. et al. [66] | 10 | Rainfall thresholds for possible landslide occurrence in Italy | Landslide information obtained from multiple sources and rainfall data captured by rain gauges to construct a catalog of rainfall events in Italy between 1996 and 2014. | 24 |
| Fabbri, A.G. et al. [67] | 9 | Favorability modeling of landslide hazard with spatial uncertainty of clab membership: a reapplication in central Slovenia | Shared the spatial database of landslide hazard prediction in central Slovenia and carried out spatial prediction experiments by verifying the technology, emphasizing the importance of the shared database. | 15 |

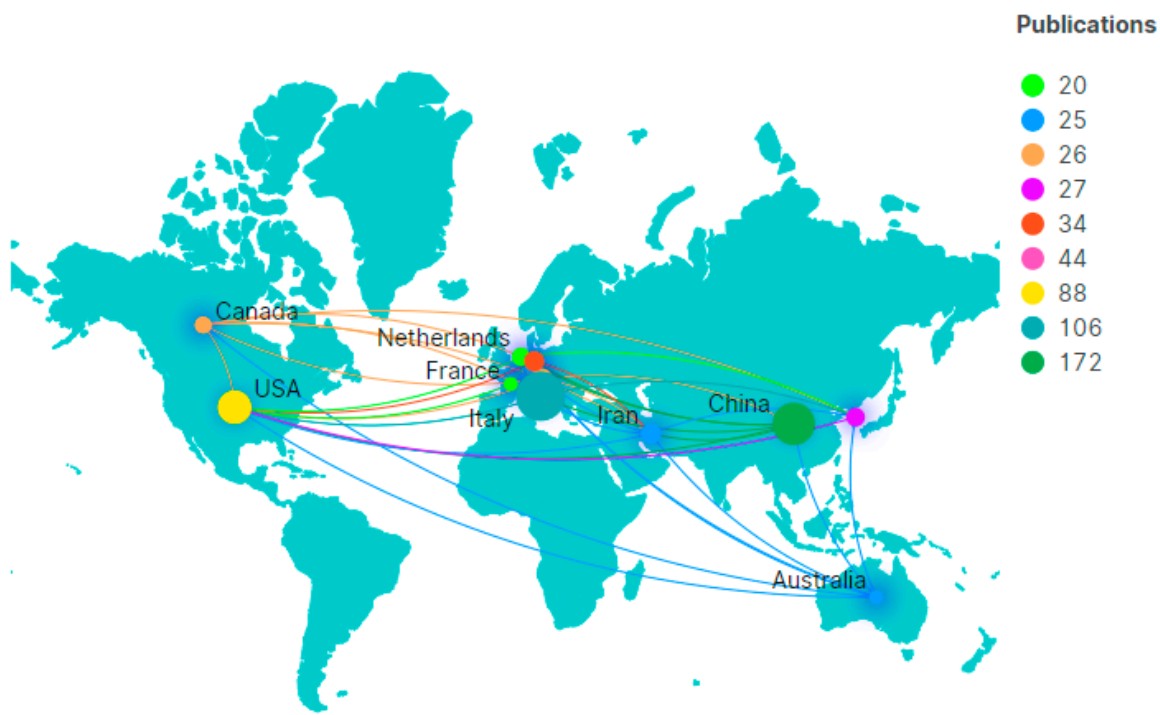

**Figure 3.** Geographic visualization map of the geographical distribution of documents.

After the data information downloaded from WOS and Scopus were synthesized, a total of 138 journals were retrieved, of which 32 journals published more than three documents, indicating that the research on the uncertainty of landslide susceptibility prediction has attracted extensive attention. Figure 4 shows the top ten journals. The journals ranking the top three in the number of documents published are ENGINEERING GEOLOGY (38), LANDSLIDES (37), and GEOMORPHOLOGY (30), with an h-index of 111, 64, and 136, respectively. As highly-ranked journals in the industry, they strongly influence engineering geology and natural geography and play an essential role in promoting the development of this field. Furthermore, the first published papers in this field in 2002, 2007, and 1996 show that the research on the uncertainty of landslide susceptibility prediction has been a concern for a long time. In particular, GEOMORPHOLOGY has been paying attention to this field for 25 years, with a higher number of documents published.

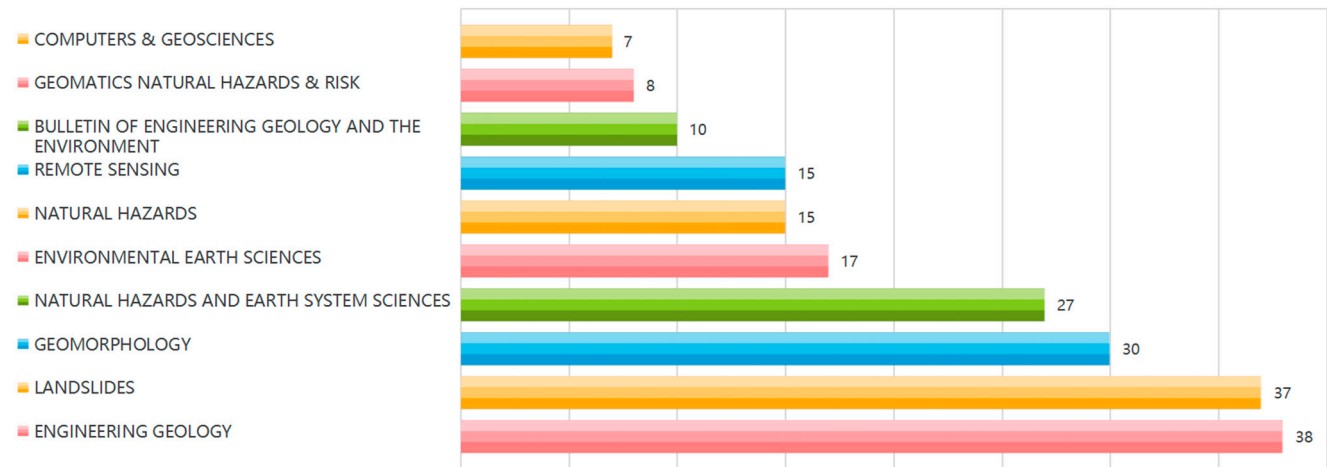

**Figure 4.** Contribution to journal publication volume (top 10).

### 3.1.3. Analysis of Highly Cited Documents

The WOS and Scopus data platforms showed that the most-cited document is a document named Estimating the Quality of Landslide Susceptibility Models published by Guzzetti F. et al. in the journal of GEOMORPHOLOGY in 2006 [63], with a number of citations of 549 on the WOS data platform and 573 on the Scopus data platform. As mentioned in Section 3.1.1, the number of documents published in the research of uncertainty in landslide susceptibility prediction had increased significantly since 2006, so Guzzetti F. et al. published this paper at the right time. This paper expounds in detail on the selection, parameter setting, establishment of prediction unit division, and other contents of the landslide susceptibility model, providing necessary enlightenment for later scholars to carry out relevant research. In addition, the five highly cited documents that rank high on both WOS and Scopus data platforms and are sorted out by screening (as shown in Table 3) showed that articles and reviews are the main types of highly cited documents. The types of documents sorted out by the document sample library in this paper are shown in Figure 5. It can be found that review, as a commonly used type of cited document, accounts for less than 1% (the number is only 14). Therefore, it is necessary to summarize and review the records in this field.

**Table 3.** Highly cited literature information table.

| References | Cite Frequency | Year | Title | Publication Type | Research Contents |
|---|---|---|---|---|---|
| Guzzetti, F. et al. [63] | 573 (Scopus) 549 (WOS) | 2006 | Estimating the Quality of Landslide Susceptibility Models | Article | The error associated with the susceptibility assessment for each mapping unit was determined by studying the variation in the model's susceptibility estimates. |
| Reichenbach, P. et al. [37] | 569 (Scopus) 257 (WOS) | 2018 | A Review of Statistically-Based Landslide Susceptibility Models | Review | Provides a critical review of statistical approaches to landslide susceptibility modeling and associated terrain zoning, provides graphical visualizations, and reveals significant heterogeneity in subject data, modeling approaches, and model evaluation criteria. |
| Gariano, S.L. et al. [30] | 419 (Scopus) 270 (WOS) | 2016 | Landslides in a Changing Climate | Review | An initial global assessment of future landslide impacts and a global map of projected impacts of climate change on landslide activity and abundance are presented. |
| Althuwaynee, O.F. et al. [68] | 247 (Scopus) 230 (WOS) | 2012 | Application of an Evidential Belief Function Model in Landslide Susceptibility Mapping | Article | Exploring potential applications of evidence belief function models in landslide susceptibility mapping using GIS. |
| Van Den Eeckhaut, M. et al. [69] | 294 (Scopus) 203 (WOS) | 2006 | Prediction of Landslide Susceptibility Using Rare Events Logistic Regression: A Case-Study in the Flemish Ardennes (Belgium) | Article | Evaluate the statistical multivariate method of rare event logistic regression to create landslide susceptibility maps. |

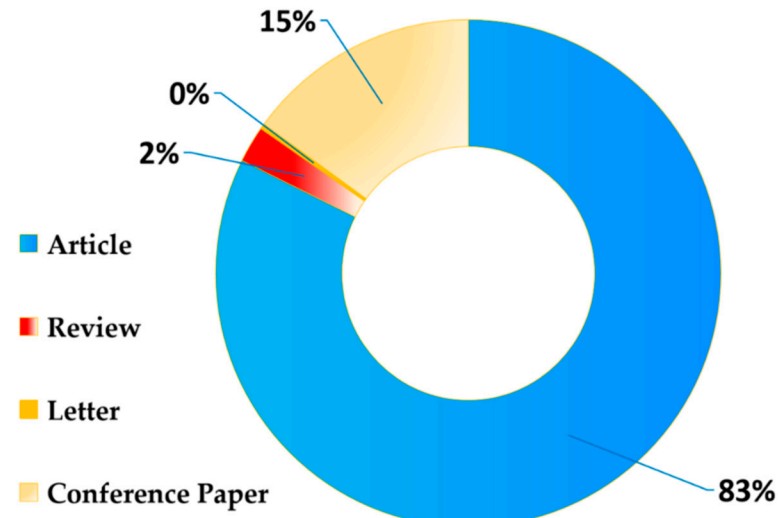

**Figure 5.** Post volume by time.

*3.2. Knowledge Mapping Analysis*

3.2.1. Analysis of Scientific Research Cooperation

In terms of the cooperation among authors, the author cooperation network knowledge mapping drawn by CiteSpace (version 5.8.R3) software is shown in Figure 6, and the authors who have published more than three documents were extracted, clearly showing that the author scientific research cooperation presents the characteristics of "scattered on the whole but locally concentrated". Three core author groups formed a locally concentrated map. According to the statistics of the number of documents published by high productivity authors in Section 3.1.2, F GUZZETTI published 16 documents, and it is the author group with the highest number of documents published in this field. The core author group is a group that is the most concentrated, has the largest number of author teams, and takes "F GUZZETTI" as the core, and its members include "STEFANO LUIGI GARIANO (12) ", "MARIA TERESA BRUNETTI (11) ", and "SILVIA PERUCCAC (10) ", which are academic teams with a large number of documents published. In this core group, the cooperation between teams is more frequent, and the contact between scholars is also closer. However, the map distance between the largest core author group and the other two larger core author groups is large, indicating that this group is relatively independent. The map distance between the larger core author group formed by the two author teams of "FAMINGHUAGN" and "THMAS BLASCHKE" and another core group composed of "JUNWEI MA" and "HUIMING TANG" is small, indicating that these two core groups have more scientific research cooperation. In addition, there are some author groups with a small number of authors but a large number of documents published. Taking the author groups of "Pradhan (8)", "CHUNG (6)", and "AG FABBRI (4)" as examples, they are relatively scattered, focus on small-scale scientific research cooperation, and they lack collaboration with larger author groups.

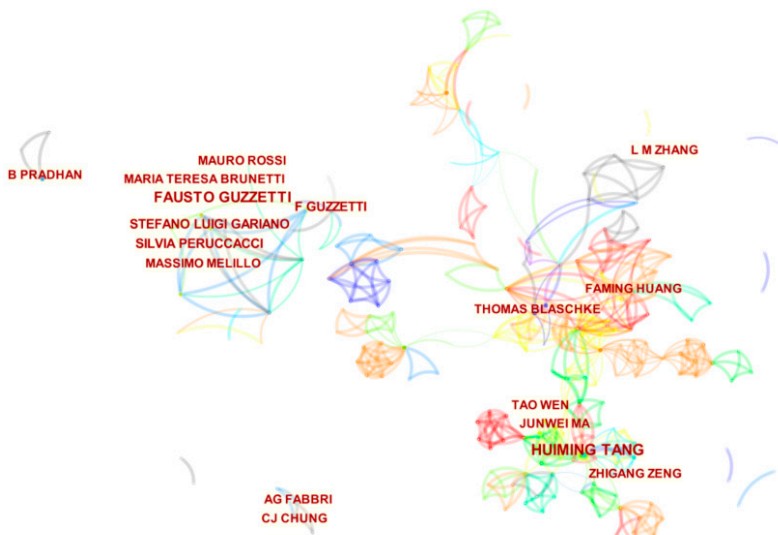

**Figure 6.** Author collaboration network knowledge graph.

To analyze the cooperation among countries, the collinear function of CiteSpace (version 5.8.R3) software was used to obtain the cooperation knowledge mapping among countries, as shown in Figure 7. The tree-ring-shaped node on the figure represents the number of documents published by the country. The larger the tree ring radius, the greater the number of published papers. The curve connection between nodes represents the scientific research cooperation between the two countries. It is not difficult to find that countries with many documents published have more academic exchanges with other countries. For example, the countries with the most significant number of papers published are China, Italy, and the United States. They have frequent cooperation with each other and

extensive scientific research cooperation with other countries. A major scientific research force group with China, Italy, and the United States as the core has been formed in this field. The economic development level of these three countries explains that the scientific research ability in this field is positively correlated with the social and economic development level to a certain extent.

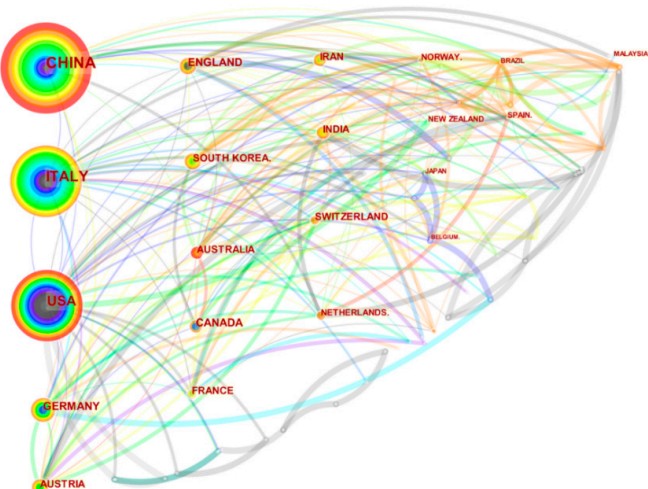

**Figure 7.** National cooperation network knowledge graph.

In terms of the cooperation among institutions, the cooperation network mapping among institutions can be obtained by the co-occurrence function of CiteSpace (version 5.8.R3) software, as shown in Figure 8. The larger the font size of the institution name on the figure, the greater the number of documents published by the institution. The higher the number of curves near the institution name, the more frequent the cooperation with other institutions. It is obvious that although there are many core institutional groups in the network, the collaboration among institutions is more frequent and closely linked. Further analysis shows that the research on landslide susceptibility prediction mainly promotes the formation of high-productivity research institutions represented by the China University Of Geosciences, Tongji University, and Sejong University. In terms of the nature of research institutions, universities, colleges, and institutes are the main research forces in this field.

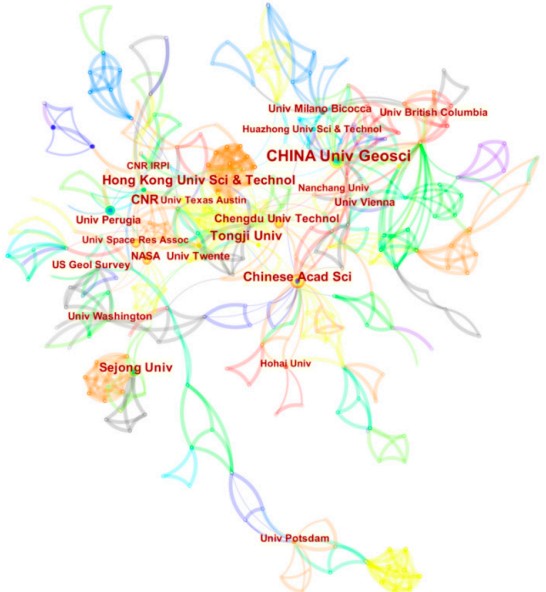

**Figure 8.** Institutional cooperation network knowledge graph.

3.2.2. Analysis of Research Hotspots

To better display hot keywords, keywords with similar meanings were first merged, then high-frequency keywords (with a frequency of more than ten times) were extracted in VOSviewer software, and keywords that were less relevant to this research were deleted for co-occurrence analysis. As shown in Figure 9, different colors represent different research topics. The larger the box area and the keyword name, the hotter the node is. The four research topics of uncertainty analysis of landslide susceptibility prediction can be summarized in Figure 9. In the green keyword group, "susceptibility" shows its status and influence as a central topic in both node location and node size. Other keywords such as "support vector machine", "random forest", and "map" focus on this central word and form a tight co-occurrence network, reflecting the high aggregation of the uncertainty research of landslide susceptibility prediction; the red keyword group takes "event" as the core and represents the research conducted focusing on the factors that cause landslides such as "earthquake", "flood" and "extreme rainfall"; the blue keyword group is relatively scattered and does not include obvious core hot keywords. However, the research content is mostly associated with the input parameters of the susceptibility prediction algorithm, which is related to the situation that the landslide prediction model has many parameters that need to be considered and also has uncertainty. The keyword group represented by yellow is more scattered and can roughly show the correlation with the early-warning index for landslide occurrence, indicating that there are many hot research directions in the research of the early-warning index of landslides.

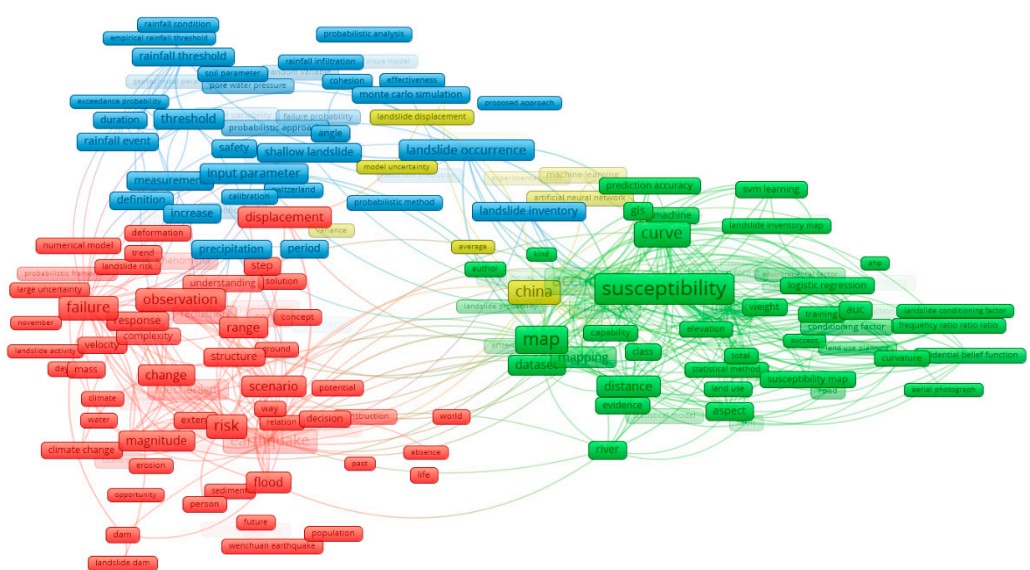

**Figure 9.** High-frequency keyword co-occurrence analysis.

To further present the relationship among research hotspots and grasp their research contents and directions, CiteSpace (version 6.1.R2) was used to draw the keyword cluster map based on VOSviewer keyword co-occurrence analysis, as shown in Figure 10. The research on the uncertainty of landslide susceptibility prediction can be summarized into 14 clusters, which are #0 debris flow, #1 lithology, #2 landslide displacement prediction, #3 support vector machine, #4 hazard assessment, #5 finite element methods, #6 slope stability, #7 Monte Carlo simulation, #8 fuzzy sets, #9 sediment, #10 malaysia, #11 catchment, #12 climate change, #13 shear strength, #14 prediction pattern. The larger the cluster number is, the greater the number of documents published of this cluster is, and the more influential it is in the research in this field. Therefore, these 14 clusters can be divided into four research topics:

(1)  Research on disaster-causing factors. It is composed of clusters #0 debris flow, #1 lithology, #6 slope stability, #9 sediment, #12 climate change, and #13 shear strength. Many factors cause landslides. The analysis results of the cluster map (Figure 10) show that the influencing factors of debris flow, underlying surface structure, and rainfall are the research hotspots.

(2)  Research on prediction units. It is composed of clusters #2 landslide displacement prediction, #10 malaysia, and #11 catchment. A representative document in cluster #2 landslide displacement prediction was published by Aydin, A. in 2006 [70], which suggests that landslide-prone slopes should be investigated on site to reduce the uncertainty of delimiting the boundary of the research zone. The three documents involving cluster #10 malaysia mainly introduce the slope failure data analysis [71] and modeling framework [72], and they reveal the uncertainty of probability prediction [73]. Although there are few documents related to this cluster, these documents were published in the past two years, indicating that this cluster's research content is a current hot research topic. Furthermore, by retrieving the abstracts of documents in the document sample database, it was found that 43 records are directly related to cluster #11 catchment, which shows that catchment is undoubtedly a hot topic of current research.

(3)  Research on data sets. It comprises clusters #4 hazard assessment and #8 fuzzy sets. From the number of groups, it can be seen that there are few research contents related to data sets. However, this does not mean scholars do not care about the topic. On the contrary, it reflects the lack of landslide data.

(4)  Research on prediction models. It comprises clusters #3 support vector machine, #5 finite element methods, #7 Monte Carlo simulation, and #14 prediction pattern. In addition to the cluster #14 prediction pattern, the other three cluster numbers related to the prediction models are highly ranked. Figure 10 intuitively shows that the three cluster color blocks cover a large area, indicating that the landslide susceptibility prediction models have attracted much attention. There are many landslide prediction models that can be selected, but it can be found that the three models of support vector machine, finite element methods, and Monte Carlo simulation are the most popular uncertainty analysis models for landslide susceptibility prediction at present.

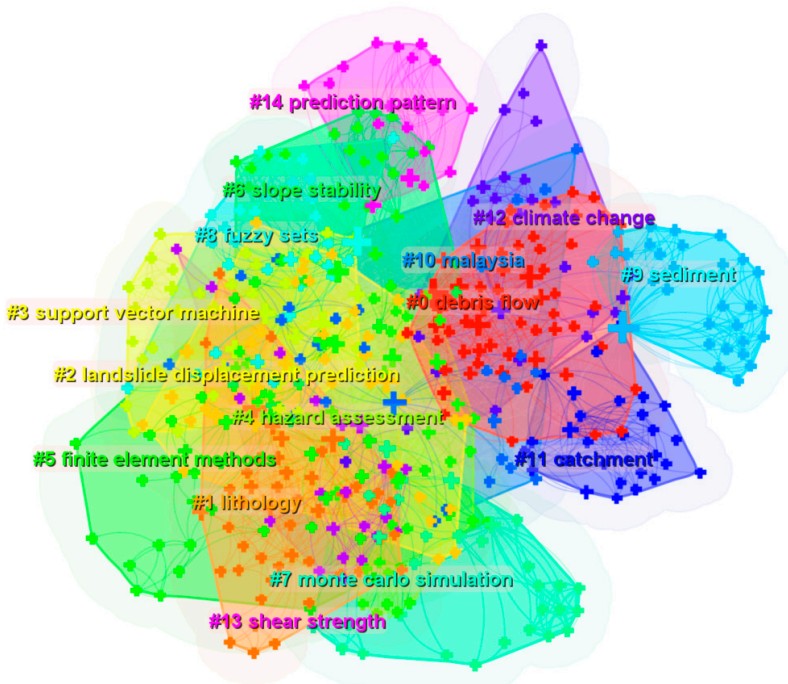

**Figure 10.** Cluster graph of Citespace.

3.2.3. Analysis of Frontier Trend

The mutation keywords refer to keywords with rapidly increased frequency in a certain period, which can be used to predict emerging trends in a particular field. The data were imported into the CiteSpace tool by CiteSpace (version 6.1.R2) for Burst detection, and the parameter settings are as follows: Gamma = 0.2, number of states = 2.0, and minimum duration = 1. The 25 mutation keywords and their durations in the uncertainty research field of landslide susceptibility prediction during 1982–2022 were obtained, as shown in Figure 11. In Figure 11, Column 1 represents keywords, and Column 2 represents the earliest years in all documents. Column 3 represents the increased intensity of the frequency of the keyword cited or the quantized value of the degree to which the keyword is concerned. The years in Columns 4 and 5 represent the keyword's earliest years and latest years, respectively. The red strip represents the duration. For example, the increased intensity value of the citation frequency of the node of the keyword flow is 2.44, the start time is 1998, and the end time is 2009; this means that the mutation keyword has been cited for 12 years and can be used to analyze the keyword mutation index in this research field and describe the frontier development trend. In order to more intuitively analyze the development process of mutation keywords in terms of time, by sorting mutation keywords in Figure 11 by the start time, it can be intuitively seen that flow appears first and lasts for the longest time, but it is not the keyword with the most incredible intensity. The keyword with the greatest intensity is uncertainty analysis, which is also the core issue of this research. Pennant Figure 12 shows that there are many factors that affect the prediction results of landslides, and it also shows that it is very difficult to carry out uncertainty analysis. In addition, although keywords such as random forest, machine learning, and neural network have appeared only in the last three years, they are not only the current research hotspot but also the research director for a long time in the future in terms of intensity and end time.

## Top 25 Keywords with the Strongest Citation Bursts

| Keywords | Year | Strength | Begin | End | 1982 — 2022 |
|---|---|---|---|---|---|
| flow | 1982 | 2.44 | **1998** | 2009 | |
| earthquake | 1982 | 2.43 | **2008** | 2015 | |
| erosion | 1982 | 3.15 | **2010** | 2013 | |
| stability | 1982 | 2.78 | **2010** | 2011 | |
| risk analysis | 1982 | 5.08 | **2011** | 2015 | |
| bayesian network | 1982 | 2.55 | **2012** | 2013 | |
| uncertainty analysis | 1982 | 5.47 | **2013** | 2016 | |
| hazard assessment | 1982 | 3.04 | **2014** | 2017 | |
| belief function | 1982 | 2.89 | **2015** | 2019 | |
| impact | 1982 | 4.01 | **2016** | 2018 | |
| frequency ratio | 1982 | 2.51 | **2016** | 2017 | |
| rainfall threshold | 1982 | 4.55 | **2017** | 2020 | |
| reliability | 1982 | 2.71 | **2017** | 2018 | |
| spatial variability | 1982 | 2.42 | **2017** | 2020 | |
| identification | 1982 | 3.28 | **2018** | 2020 | |
| logistic regression | 1982 | 2.54 | **2018** | 2020 | |
| induced shallow landslide | 1982 | 2.42 | **2018** | 2019 | |
| random forest | 1982 | 3.55 | **2019** | 2022 | |
| extreme learning machine | 1982 | 3.06 | **2019** | 2020 | |
| land use | 1982 | 2.7 | **2019** | 2020 | |
| spatial prediction | 1982 | 2.67 | **2019** | 2020 | |
| 3 gorges reservoir | 1982 | 2.46 | **2019** | 2022 | |
| machine learning | 1982 | 4.67 | **2020** | 2022 | |
| decision tree | 1982 | 3.26 | **2020** | 2022 | |
| neural network | 1982 | 2.56 | **2020** | 2022 | |

**Figure 11.** Keyword mutation analysis.

The timeline chart analysis was made by CiteSpace (version 6.1.R2) for keywords in the research field of the uncertainty of landslide susceptibility prediction, and the evolution routes of all clustering keywords are shown in Figure 13. To observe the evolution route of keywords, timeline chart 13 adopted the same LSI algorithm as clustering map 11. Therefore, the keyword clustering names in timeline chart 13 are consistent with Figure 10. In Figure 13, a ring-shaped tree node on the timeline represents a keyword. The larger the tree ring diameter is, the more frequently the keyword appears. The solid-line part on each timeline represents the duration of this cluster. The #9 sediment is a firstly researched cluster,

and its research began in 1995 and ended in 2009, lasting only 15 years, while #0 debris flow has the most extended duration of 26 years, and its research began in 1996 and is still being researched. The continued clusters include #2 landslide displacement prediction, #3 support vector machine, #5 finite element methods, #7 Monte Carlo simulation, #10 malaysia, #11 catchment, and #12 climate change. Although these clusters started a little later, they all represent the frontier research trend in the future.

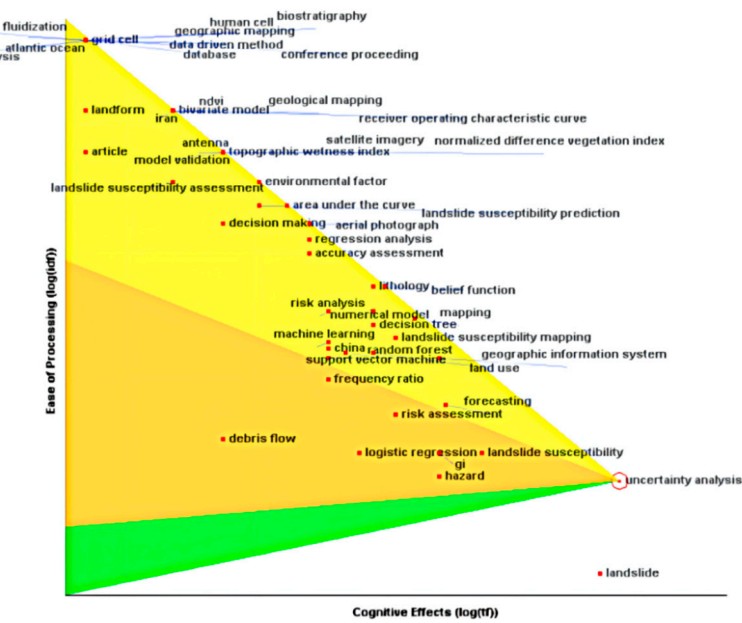

**Figure 12.** Uncertainty analysis pennant plot.

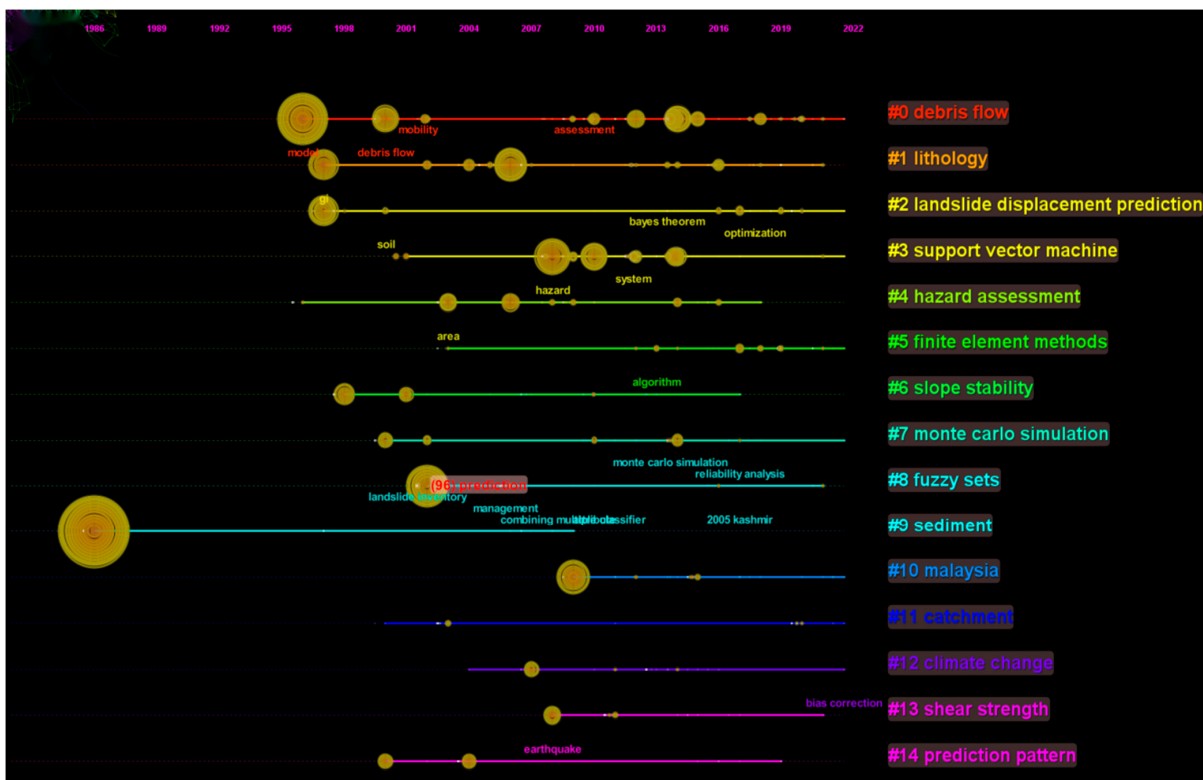

**Figure 13.** Timeline diagram based on LSR algorithm.

## 4. Main Research Subfields

There are many uncertain factors in the modeling process of landslide susceptibility prediction [74], such as the accuracy of topographic data [75], the selection of environmental factors [76] and their spatial resolution [77], the choice of non-landslide samples [78], the proportion of model training test set [79], and the determination of prediction models [80]. A landslide geological model is a simple expression that abstracts and generalizes the main engineering geological elements and deformation and failure of landslides through a comprehensive analysis of engineering geology based on landslide characteristics. The main reasons for the uncertainty in this process are as follows:

(1) Regarding objective factors, the landslide system is influenced by random conditions and processes. The internal conditions cause the influence, external factors, and the interaction of internal and external factors that constitute the landslide system. When the landslide disaster prediction models are built, the data used often come from a small number of the known and observable key influencing factors, but a large number of information that is unknown or difficult to obtain is not taken into account, which makes the landslide prediction model only approximate to the actual situation in essence and increases the uncertainty of the models [81].

(2) In terms of the influence of subjective factors, the process of building landslide prediction models is easily influenced and interfered with by the subjective factor of the human cognitive level and the lack of accurate understanding of the prediction unit, data set selection, and model determination of landslide prediction models will inevitably greatly increase the uncertainty of the model [82].

It is clear that the modeling process of landslide susceptibility prediction is bound to be affected by the above factors, resulting in a certain degree of uncertainty in the prediction results of the model. In combination with the results of bibliometric analysis and knowledge mapping analysis in Section 3 of this paper, it was found that the research on the uncertainty of landslide susceptibility prediction first needs to establish a correct understanding of the impact of disaster-causing factors on landslides and the nonlinear correlation between disaster-causing factors, then correctly divide the prediction units, select the original data set from the massive monitoring data, and decompose the data set into the training set and test set in proper proportions, to lay the foundation for the correct selection of models and reduce the uncertainty of landslide disaster prediction results. Therefore, this paper divided the research on the uncertainty of landslide susceptibility prediction into the following four subfields for discussion: (1) uncertainty analysis of disaster-causing factors, (2) uncertainty analysis of prediction units, (3) uncertainty analysis of model space data sets, and (4) uncertainty analysis of prediction models, as shown in Figure 14.

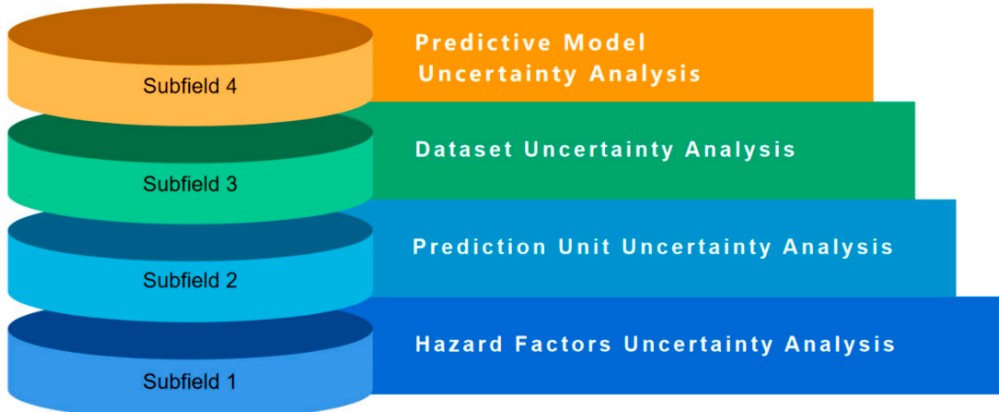

**Figure 14.** Subfield composition.

### 4.1. Uncertainty Analysis of Disaster-Causing Factors

The occurrence of a landslide disaster is a complicated nonlinear dynamical system process with uncertainty [83], in which there are often many hidden and uncertain factors [84]. Its evolution process is affected by many factors, such as structure, rainfall, landform, and human activities, and it is also the result of the combined effect of various factors [85]. The landslides cannot be comprehensively analyzed and predicted without any of them. From a macroscopic point of view, these impact factors can be divided into two categories: trigger factors and environmental factors; as shown in Figure 15, a color represents a class of influencing factors. Three main factors trigger landslides [7]; the first trigger factor is the impact of earthquakes. Massive landslides are often triggered by earthquakes [86], and the disasters caused are also huge; the second trigger factor is the actions of water. Continuous rainfall [87] and snow melting [88] will saturate the soil and reduce the lubrication friction coefficient of the sliding surface [89], resulting in landslides. The third trigger factor is human activities [90]. Unreasonable human excavation will destroy the conditions of the underlying surface, especially human activities such as illegal mining tree cutting, reducing the earth's regulation capacity and creating conditions conducive to landslides. Many landslide events show that the probability of landslides caused by human factors may be greater than that caused by climatic factors. By summarizing the landslide influencing factors used in the previous literature, the environmental factors that cause landslides mainly include geological environment [91], topography environment [4], and hydrological environment [92]. Among them, slope is the most considered factor by most literature [93]. Obviously, the slope is the initial factor that needs to be considered in the uncertainty analysis of landslide susceptibility prediction.

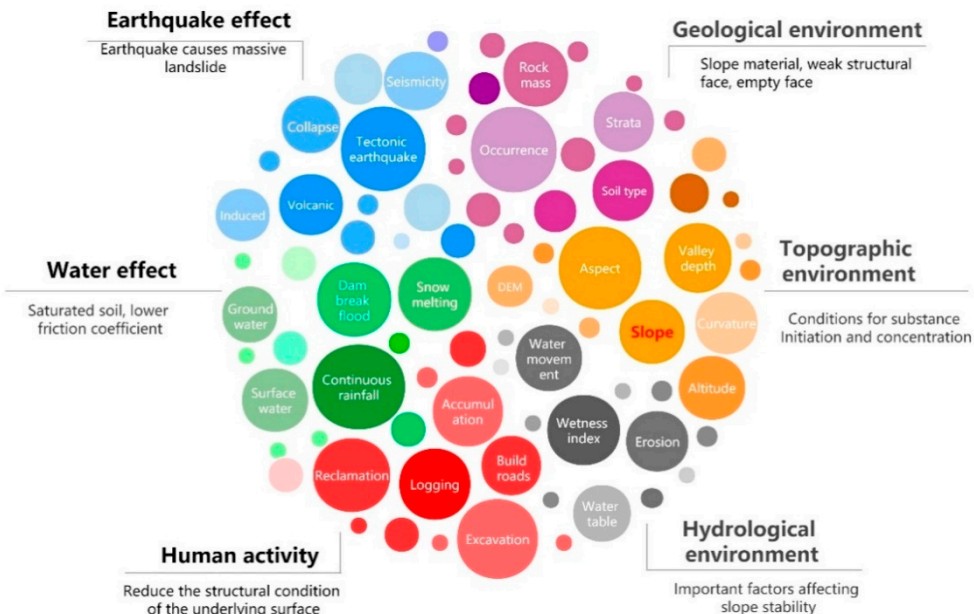

**Figure 15.** Landslide influencing factors commonly considered in the literature.

The basic work of landslide disaster prediction is to analyze the impact of various disaster-causing factors on the occurrence of landslides and the potential combined influence of various disaster-causing factors [94]. As there is a specific correlation between the various index factors that affect the formation of landslides, these index factors were analyzed to reduce the superimposed impacts between them [95]. The landslide disaster-causing factors often have the characteristics of uncertainty and complexity, and it is generally difficult to quantitatively analyze the effect of these factors and predict the probability of occurrence of landslide disasters. According to the nonlinear correlation analysis of disaster-causing impact conditions and characteristics based on statistical analysis theory,

landslide prediction can be connected with disaster-causing factors for qualitative analysis. The commonly used connection methods include the weight of evidence, information entropy, frequency ratio, probability statistics, etc. [96]. As a geo-statistical method, the weight of evidence makes a superimposed compound analysis of some geographic information related to the formation of landslide disaster through Bayesian statistical analysis mode to predict dangerous areas. It is an organic combination of mathematical statistics, image analysis, and artificial intelligence and also provides an effective way for landslide disaster prediction based on the GIS platform. Information entropy is the basic concept of information theory, which mainly describes the uncertainty of various possible disaster events in landslide information sources. The frequency-ratio method is simple in structure and can effectively reflect the effect of environmental factors on the probability of occurrence of landslides. Under the premise of probability axioms, probability statistics believe that, although a variety of landslide events may occur in a random sampling, the landslide disaster event (outcome) that is most likely to occur (encounter) is the event with the highest probability. Currently, there is no unified demonstration or empirical basis for determining which connection method to use for the nonlinear correlation analysis of landslide disaster-causing factors. Table 4 shows several typical cases in the documents.

**Table 4.** Typical cases of nonlinear correlation analysis of disaster-causing factors.

| References | Purpose | Factors | Research Idea | Conclusion |
|---|---|---|---|---|
| Tsai, F. et al. [97] | Verify landslides caused by heavy rainfall in Taiwan | Topographic and vegetation factors | Used decision tree and Bayesian network data mining algorithm to extract landslide factors from the provided knowledge and develop a statistical-based mechanism to reduce data uncertainty. | Apply the model to landslide prediction directly, and the prediction results will be unreliable due to the spatial uncertainty of data. |
| Tsai, T.L. et al. [98] | Evaluate shallow landslides caused by rainfall | Soil parameters, slope conditions, and hydrological conditions | Compared the applicability of Rosenblueth point estimation method and Monte Carlo simulation method by using various soil parameters, slope conditions, and hydrological conditions. | The correlation of soil parameters reduces the safety standard deviation factor but does not affect the safety mean factor. The prediction error may occur if the correlation of soil parameters is ignored. |
| Wang, X. et al. [99] | Analyze the correlation between relevant factors and landslide | Lithology, relative relief, tectonic fault density, rainfall, and road density | Proposed using the CLSI and CLAI calculated based on the frequency ratio to express the correlation between various factors and landslide occurrences. | CLSI is helpful to reduce the uncertainty of sensitivity assessment when the landslide inventory with non-uniformity problems is used. |
| Oguz, E.A. et al. [100] | Quantify the impact of disaster-causing factors on prediction uncertainty | Geotechnical and hydrological parameters | Developed a three-dimensional slope stability model combined with random field model and Monte Carlo method to capture spatial variability and predict landslide susceptibility. | The new model has higher landslide prediction accuracy than the traditional model. |
| Lian, C. et al. [101] | Affect the interaction of different inducing factors of landslide evolution | Structure, rainfall, and reservoir water level fluctuation | Proposed a new neural network technology of extreme learning machine integration (E-ELM) and used the grey correlation analysis (GRA) method to screen out the inducing factors with great influence as the input factors in E-ELM. | The model can predict the trend component displacement and periodic component displacement, and the total predicted displacement is obtained by adding the predicted displacement values of each factor. |
| Huang, F. et al. [102] | Study the influence of environmental factor attribute interval division quantity on modeling | Topography and landform, formation lithology, hydrological environment, and surface coverage | Obtained the landslide inventory and its environmental factors in the research area and carried out frequency ratio analysis on continuous environmental factors under the condition of dividing quantitative values in different attribute intervals. | When the frequency ratio analysis is made, there is a critical point in the continuous environmental factor of the landslide that can effectively avoid too complex of a frequency ratio calculation, while ensuring the prediction accuracy. |

It is difficult to identify the relationship between the factors that can control and affect the preparation and occurrence of landslides and the relationship between factors and the occurrence of landslides. The feature extraction and nonlinear correlation analysis of disaster-causing factors is complex and challenging research. In addition, due to the lack of historical data required to determine the occurrence of landslides, a large number of research needs to be conducted to reduce the uncertainty of disaster-causing factors that affect landslide prediction. Therefore, scholars rarely use a single method to solve such complex problems but mostly analyze the fate of disaster-causing factors by integrating a variety of analysis methods. For example, some scholars combine the soil moisture routing (SMR) model and the infinite slope model with probability analysis by grid-based tools to analyze the relationship between disaster-causing factors such as climate, digital elevation model, soil and land use, and disaster-prone areas [103]; some scholars create the optimal landslide sensitivity zoning map by weight of evidence, landslide frequency ratio, and fuzzy logic methods [95]. The weight of evidence is used for continuous classification factor data, the landslide frequency ratio assigns a membership degree to factor category, and the fuzzy logic method integrates the membership values; other scholars determine the landslide location and disaster-causing factors by the GIS-based frequency ratio (FR) statistical method [104] and the multi-criteria decision-making (MCDM) [105] and hybrid SMCE integrated method [106]. Each method used for the uncertainty analysis of disaster-causing factors has its advantages and disadvantages. Therefore, there is no absolute optimal method, only the method most suitable for the landslide prediction effect. Although it is difficult for scholars to clearly and reasonably explain the reasons for choosing this analysis method, a large number of practices have proved that the uncertainty of landslide disaster prediction calculated by a single method (due to its incomplete function and poor adaptability) is often higher than the prediction results calculated by integrating multiple methods. In short, complex nonlinear problems should be handled by integrated methods.

### 4.2. Uncertainty Analysis of Prediction Units

The prediction of landslide disasters is based on the division of prediction units in the research areas. According to the specific geological and topographic conditions in different regions, the corresponding division methods are used for disaster-causing factors to divide the shape and size of prediction units and further carry out the uncertainty analysis of the model space data sets, prediction models, and prediction results in combination with the uncertainty analysis of disaster-causing factors. In the research process, the researchers analyze the distribution of disaster-causing factors and landslide disasters in the research area, divide the prediction units based on the screened disaster-causing factors, and divide the secondary disaster-causing factors by reasonable unit division methods. The division methods can be roughly divided into two categories: regular units and irregular units. Some more representative case studies are shown in Table 5.

The reasonable division of prediction units plays a certain role in optimizing the prediction model, reducing the uncertainty of landslide disaster prediction, and improving the prediction effect. Although there is no reasonable explanation to determine which division method is more advantageous at present, the cases in the documents show that the regular grid unit division method is less used in the two ways. Although the grid division method is simple, it is highly subjective. It often has problems of too many or too few grids and artificial destruction of the integrity of landslide prediction, leading to the distortion of prediction results. In most cases, the irregular unit division method is used. Although this method is more complex and uses polygons with different shapes, it has a clearer geological significance and is more in line with the irregularity of landslide boundaries. In particular, with the wide application of GIS technology, the analysis, calculation, and drawing ability of landslide prediction has been greatly improved, which can not only realize the vector superposition of any layer to solve the division problem of the prediction unit but also does not need to take into account the errors and mapping difficulties caused by boundary differences.

**Table 5.** Typical cases of nonlinear correlation analysis of disaster-causing factors.

| Division Method | Research Content | Object to Be Divided | Method | Conclusion |
|---|---|---|---|---|
| Regular units | Obtain the optimal landslide sensitivity assessment [107] | Lithology and soil hydrological information in regional landslide inventory | Divide the research area into grid units and topographical units, subdivide the slope units according to the topographic gradient to obtain hydrological morphological units, and determine a single pixel as the representative of the landslide depletion area for grid units. | It minimized the inherent limitations of regional landslide inventory and sensitivity maps. |
| | Influence of soil depth on the probability of occurrence of landslides [108] | Soil characteristics and vegetation classification | Develop a source tracking algorithm (STA), and use the spatial variable supplementary data from a hydrologic source domain (HSD) and the spatial distribution grid data of soil characteristics and vegetation classification to characterize the parameter estimation of the probability distribution of the model input uncertainty. | "Over-representation" areas that suffer from shallow landslides may be misleading. Locations with high landslide probability other than landslides can be used as the index for the additional investigation of missing areas. |
| | Evaluate the spatial variability and uncertainty of model parameters [109] | Slope, soil strength | Analyze the grid data, provide the estimated value of parameter area with relevant error range by the Kriging method and display the safety factor calculated at each point of the grid to identify the landslide. | This landslide prediction method can be improved by using the Kriging method. |
| Irregular unit division | Number of landslide events when rainfall reaches its threshold [110] | Landslide inventory, daily rainfall, and effective cumulative rainfall | Divide the research area into multiple rain gauge control areas by the improved Thiessen Polygon method and divide the control area into slope units that reflect the topographic characteristics to improve the spatial resolution of the rain gauge. | The rainfall of at least one rainfall event in the slope unit exceeds the threshold. |
| | Uncertainty of different landslide boundaries on modeling [76] | Slope, lithology, and other environmental factors | Establish the correlation between the landslide boundary and the frequency ratio of the landslide boundary and environmental factors based on landslide points, buffer circles, and polygonal surfaces, and then select multilayer perceptron and random forest to build the model. | Compare the method using a polygon surface with the method based on point and circle, and the boundary and spatial shape can significantly improve the accuracy of landslide sensitivity map LSM. |
| | Predict the scope of landslide [111] | Plane area of landslide | First introduce a statistic-based model, establish models on the slope triggered by the landslide in response to seismic vibration, and simulate the expected failure surface on the slope without landslide. | The model can estimate the plane area of the landslide aggregated by each slope unit. |
| | Evaluate the influence of topographic mapping unit on data-driven landslide sensitivity map [112] | Slope angle, aspect, slope area ratio, lithology, and land use / land cover | Calculate the landslide sensitivity model by using the same topographic mapping unit (slope topographic unit) and the complete landslide inventory represented by polygon features. | The accuracy of landslide spatial location is the key criterion for selecting the most suitable topographical unit for modeling; when the spatial accuracy is low, the grid topographic units should not be used, and the use of irregular units can help to reduce the adverse effects caused by location errors. |

*4.3. Uncertainty Analysis of Model Space Data Sets*

Landslide prediction needs to be supported by many sample data so that the prediction results can be more reliable. Data sampling/collection processes are crucial for landslide susceptibility mapping. By providing more field data, the accuracy and reliability of landslide susceptibility mapping can be improved, thereby reducing uncertainty [113]. With the development of geographic information technology, information and communication technology (ICT), it has become possible to use citizen science (CitSci) methods in the landslide data collection process [114], which has huge advantages in landslide data collection, validation, and interpretation potential, thereby contributing to the study of landslide prediction uncertainty. Since Goodchild, M.F. et al. coined the term Volunteer Geographic Information (VGI) in 2007 [115], public awareness of landslides has continued to improve, and people are not only consumers but also collectors of landslide data [116], even willing to actively participate in the process of risk and disaster management [117]. A single data set is usually divided into the training and test sets for landslide prediction and accuracy verification [118]. Given the diversity of data sets and processing methods, it is important to determine the optimal combination by comparing the results of data sets obtained by different methods [119]. Although there is no consensus between the optimal data set and the evaluation method, some data sets (such as slope angle, lithology, land use/coverage, etc.) have been widely accepted as the basis of landslide prediction. Data sets usually correspond to specific landslide events and historical lists (such as slope angle, aspect, height, plane curvature, profile curvature, dynamic river index, sediment transport index, topographic wetness index, distance to the river, distance to road, distance to fault, NDVI, land use, lithology, rainfall, and other data) [118]. Thanks to the development of various landslide disaster prediction technologies, more and more landslide prediction data are available worldwide with different accuracies. These monitoring data are various characteristic variables related to the landslide state and inevitably contain some errors. The main reasons for the uncertainty of the accuracy of landslide prediction data sets include imperfect measuring systems or instruments, restricted technical means, disturbed environmental conditions, etc. The common landslide disaster prediction data and application cases are shown in Table 6.

**Table 6.** Landslide disaster prediction data and application cases.

| Location | Data | Application | Conclusion |
|---|---|---|---|
| Countries of the European Community [120] | Remote sensing landslide data | Emphasize the image types required for different analysis scales; monitor the activities of existing landslides by GPS, photogrammetric technology, and radar interferometry; spatio-temporal analysis and prediction of slope failure | Integrate remote sensing technology into the overall framework of landslide prediction uncertainty analysis technology. |
| The Cascade Range in western Oregon, USA [121] | DEM and geological data | Process the DEM to generate a series of slope stability maps and evaluate the uncertain impact of elevation error on landslide sensitivity | The evaluation of the ability of uncertainty may help to understand the advantages and disadvantages of digital data and spatial information system applications. |
| Kaikoura, New Zealand [122] | Multi-temporal airborne laser radar data set | Propose to use a new semi-automatic 3D point cloud difference method to detect the landform variations, filter the false landslide detection caused by laser radar elevation error, obtain a robust landslide list with uncertainty measurement, and directly measure the volume and geometric characteristics of the landslide. | The size dependence detection of the system is insufficient in the 2D list, while the 3D derived list can be used to detect various hillside movements that cannot be captured by 2D landslide surveying and mapping. |

Table 6. *Cont.*

| Location | Data | Application | Conclusion |
|---|---|---|---|
| Italy [123] | FraneItalia database | Use thousands of landslide events for the reanalysis of uncertain data sets to obtain precipitation and volumetric soil moisture data | Compared with the reanalyzed soil moisture data, precipitation information is still a better prediction index to trigger landslides. |
| Switzerland [124] | Soil moisture and rainfall data | Propose a sequential threshold method, which first is divided into dry and wet preconditions through antecedent soil saturation threshold, and then estimate two threshold curves with different total rainfall duration. | The combination of soil moisture state estimation and infinite slope method can improve the separation between triggered and non-triggered rainfall events of landslides. |
| Northwestern Tunisia [125] | Data of landslide regulating factors such as elevation, slope, and aspect | Draw landslide sensitivity maps by two bivariate statistical models (evidence belief function (EBF) and weight of evidence (woe)), and landslide inventory maps by aerial photos, satellite images, and field surveys. | The landslide sensitivity maps of the two models are very similar, but the WoE model is more effective and can be used for the future planning of the research area. |
| Oregon [126] | Laser radar derived data set | Make use of the laser radar derived data set and set up the research area through several widely used statistical technologies to realize landslide sensitivity analysis | Only a few factors are needed to produce a satisfactory ability sensitivity map with high predictability. |

The construction of landslide inventories is also very important for landslide susceptibility mapping. Studies have shown that the error of landslide prediction susceptibility mapping mainly comes from landslide inventory, and the lack of detailed landslide inventories often limits the uncertainty analysis in landslide prediction [127]. For example, the construction of landslide inventories is challenging due to the lack of access to high-altitude areas, for which interpretation and heuristics of remote sensing datasets are combined with statistical sensitivity models to overcome the limited spatial coverage [128]. On the other hand, most landslide inventories are not updated over time and therefore may not capture the effects of climate, land-use change, etc. The construction of these landslide inventories requires extensive field or remote sensing work, and the construction of landslide inventories based on citizen reports has the potential to overcome these limitations. This landslide inventory is developed through the establishment of a national database, and citizens report remote sensing information to online systems so that landslide information can be updated and recorded in real time, resulting in improved landslide susceptibility predictions at a lower cost and a higher resolution accuracy [129]. Although this type of landslide inventory is promising, further development of normative data standards is needed due to possible spatial uncertainty and reporting bias in the data.

High-quality input data can improve landslide susceptibility mapping and improve prediction accuracy [130]. In order to improve the quality of input data, spatial data cleaning is crucial to remove uncertainty-type information such as ambiguity, noisy data, inconsistency, etc., hidden in the input spatial dataset. Although data fusion can be used in preprocessing to deal with the problem of data ambiguity, band-pass or slot filtering is used to eliminate periodic noise, Fourier transform is used for filtering to eliminate spike noise, and resampling is used in a certain amount of time to some extent solve the problem of inconsistency, but overall there is no general method for spatial data cleaning. Hsu, P.H. et al. attempted to use kriging interpolation to calibrate quantitative rainfall data with rainfall observations from rain gauge stations to eliminate data inconsistencies [131]. The results show that after data cleaning, Kriging interpolation-based methods can effectively correct inconsistencies, and artificial neural network analysis algorithms (ANNs) are applied to integrate large amounts of spatial data collected from historical risk events, dynamic real-time such as rainfall and water levels. The uncertainty of observations and the results of

different risk models can make landslide predictions accurate to 92.3%. Using spatial data mining and knowledge discovery (SDMKD) technology, landslides caused by rainstorms can be effectively mapped from remote sensing images and geospatial data.

A model space data set usually decomposes the original data set into the training set and test set, with different proportions, and the flow chart is roughly shown in Figure 16. The choice of model parameters in landslide susceptibility mapping makes a major determinant of model accuracy [132]. With regard to machine learning models, the determination of their hyper-parameters are crucial. Usually, hyperparameters need to be optimized, and a set of optimal hyperparameters is selected to improve the performance and effect of machine learning. Early research literature on landslide susceptibility mapping models was more focused on the comparison of the modeling accuracy of different methods, rather than the application of hyperparameter optimization in landslide machine learning modeling [133]. Later, people gradually realized the importance of hyperparameter optimization and carried out some optimization methods about hyperparameters. For example, in 2010, Wan, Z. et al. proposed a simple, practical, and time-efficient method for selecting hyperparameters for orthogonal design [134]. In 2014, Wang, X. et al. proposed a Gaussian kernel-based hyperparameter selection method for SVM [135]. In 2020, Sun, D. et al. used a Bayesian optimization algorithm to optimize hyperparameters and established some high-precision random forest landslide sensitivity evaluation models [133].

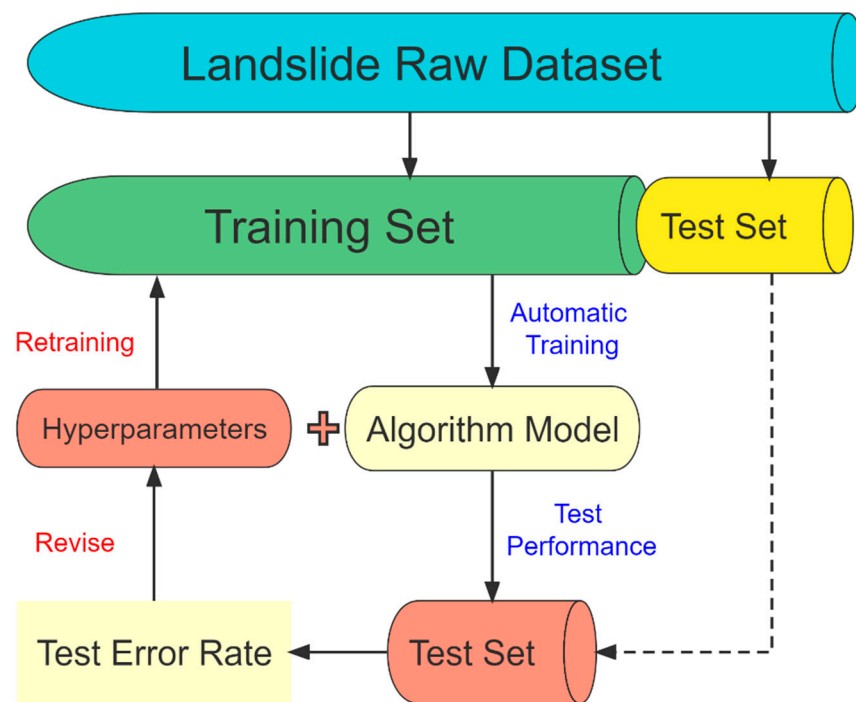

**Figure 16.** Computational flowchart for model space datasets.

At present, many kinds of methods and models have been widely used in the prediction of landslide disasters, but one of the difficulties in the application of these methods and models is the uncertainty of the model space data set. The spatial variability, measurement error, uneven density distribution, incomplete information, and other characteristics of data sets will cause the uncertainty of input parameters and the analysis process. For this reason, people continue to explore measures that can compensate for the lack of data. For example, to deal with uncertainty propagation through physical models, the landslide sensitivity is evaluated by combining fuzzy theory with the vertex method and point estimation. The results show that the fuzzy method can appropriately respond to the landslide sensitivity analysis based on physical uncertainty in watersheds [38,136]. Furthermore, the physically-based TRIGRS model has been successfully applied to evaluate rainfall-induced shallow

landslides in different research worldwide [137]. In particular, the San Carlos, Colombia application shows that TRIGRS can become a valuable tool for landslide disaster prediction in tropical mountainous areas without data.

### 4.4. Uncertainty Analysis of Prediction Models

In terms of the time-space relationship, landslide prediction models can be divided into two categories: space and time. To sum up, the space prediction model mainly includes the information model, statistical model, expert system prediction model, grey system model, pattern recognition model, and nonlinear model. In contrast, the time prediction model mainly includes a long-time prediction model (grey catastrophe model) and critical sliding time prediction model (Verhulst grey model based on displacement information, friction heat information model based on the sliding surface, landslide dynamic displacement prediction model based on rainfall process). In the calculation process of landslide disaster prediction, careful prediction models are fundamental because the wrong selection of prediction models and setting model parameters will lead to strong uncertainty in the landslide prediction results. According to the document sample library analysis in Chapter 3 of this paper, statistical and pattern recognition models have been widely used in the uncertainty analysis of landslide disaster prediction. In addition, as the two leading space models, they are the key contents of this subfield.

### 4.4.1. Statistical Model

The statistical model is used to establish the correlation relationship between landslide disasters and various factors or the combination relationship between landslide-induced factors by studying the statistical law between geological and geomorphic environmental factors, in the places where the existing landslide disasters and similar unstable phenomena occur, and inducing factors, to build the corresponding prediction model. As shown in Figure 17, the landslide disaster prediction models that have been actively researched mainly include the regression prediction model, discriminant analysis model, cluster analysis model, etc. Typical cases of statistical models in landslide disaster prediction are shown in Table 7.

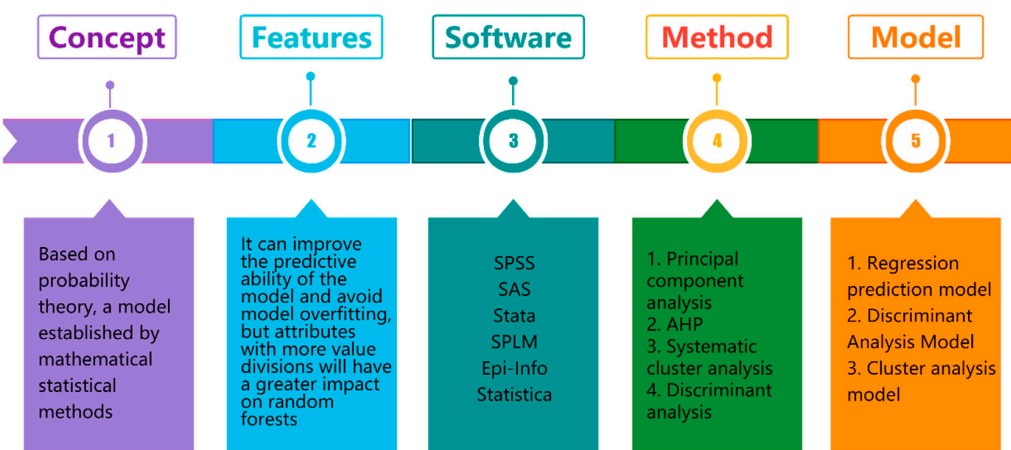

**Figure 17.** Statistical model.

**Table 7.** Typical cases of statistical models in landslide disaster prediction.

| References | Research Content | Model Used | Method | Conclusion |
|---|---|---|---|---|
| Thiery, Y. et al. [138] | Test of the performance of statistical evaluation method for landslide sensitivity | Weight of evidence (WOE) | Identify variables that represent the optimal response; test evaluate the simulation performance by the optimal combination of predictive variables and new predictive variables; evaluate the statistical model through expert judgment | The bivariate method can effectively evaluate the landslide sensitivity at the scale of 1:10,000. When RV and PV are complex or limited by an insufficient amount of information, expert knowledge needs to be introduced into the statistical model to generate a reliable landslide sensitivity map. |
| Schicker R. et al. [139] | Extraction of landslide inventory data from the original database | Logistic regression and weight of evidence | Use sensitivity maps that are predicted successfully and derived and evaluated by ROC curves for logistic regression and weight of evidence | The WOE method cannot successfully predict landslides other than the original data. |
| Rossi, M. et al. [140] | Evaluation of landslide sensitivity | Support vector machine, logistic regression | Describe the structure of software for landslide sensitivity evaluation, explain the input and output, and illustrate the specific applications with maps and graphics | Complete and comprehensive landslide sensitivity evaluation includes a model performance analysis, prediction skill evaluation, and error and uncertainty estimation. |
| Shepheard, C.J. et al. [141] | Variation of rock and soil parameters | Regression analysis | Determine the possible range of slope safety factor and the relative influence of other rock and soil parameters (such as topsoil depth and rainfall) through the statistical analysis combined with numerical simulation | A database of particle size distribution, in-situ moisture content, Atterberg, and direct shear box test results was set up. |

### 4.4.2. Pattern Recognition Model

A pattern recognition model is a model based on artificial intelligence technology, which simulates the function of human beings to perceive the outside world by the mode of replacing or helping human beings to perceive with computers. With the continuous development of artificial intelligence, machine learning methods have become increasingly popular in recent years [37]. Machine learning models have broad application prospects in predicting landslide disasters due to their advantage of high prediction accuracy [142]. At present, the prediction model of machine learning is mainly based on five algorithms, including an ensemble learning algorithm, interpretation algorithm, clustering algorithm, dimension reduction algorithm, and similarity algorithm, as shown in Figure 18.

(1) The ensemble learning algorithm is mainly used in regression and classification or supervised learning problems. Due to its inherent properties, the ensemble learning algorithm is superior to all traditional machine learning algorithms, including Naïve Bayes, SVM, and decision tree.

(2) Interpretation algorithm: It can identify and understand variables with statistically significant results. Therefore, instead of creating models to predict landslide disasters, it is better to develop interpretative models to understand the relationship between variables in the landslide disaster prediction models, including the SHAP algorithm and LIME algorithm.

(3) Clustering algorithm: It is an unsupervised learning task used for clustering analysis, which usually groups the data into clusters. Unlike the known target variables of supervised learning, there are traditionally no target variables in a clustering analysis.

A clustering algorithm can be used to find natural patterns and trends of landslide displacement data. It includes K-means clustering and hierarchical clustering.

(4)     Dimension reduction algorithm: It is a technology used to reduce the number of input variables (or characteristic variables) in the data set. With the increase in dimension (the number of input variables), the volume of spatial data of the landslide model increases exponentially, which eventually leads to the sparse data of main control factors for landslide prediction. It includes the principal component analysis (PCA) and linear discriminant analysis (LDA).

(5)     Similarity algorithm: It refers to those algorithms used to calculate the similarity of records/nodes/data points/text pairs and includes the similarity algorithm for comparing the distance between two data points (such as Euclidean distance) and the similarity algorithm for calculating text similarity (such as the Levenshtein algorithm).

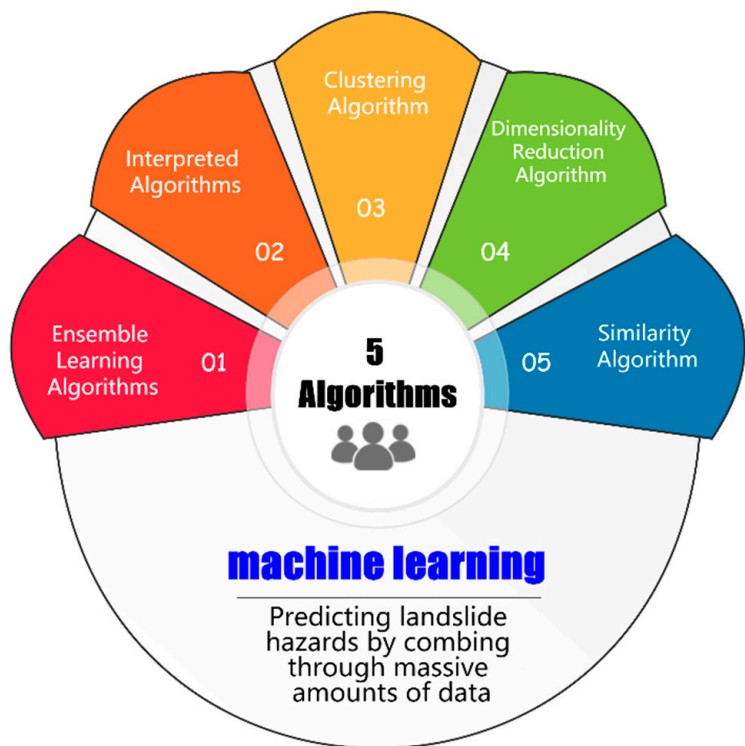

**Figure 18.** 5 Algorithms of machine learning.

There are many machine learning models for landslide disaster prediction [143], including BP neural network, multilayer perceptron, fuzzy mathematics, support vector machine, decision tree, random forest, various integration models, and recently developed deep learning models [144]. Table 8 shows several typical cases of the application of machine learning in landslide disaster prediction.

**Table 8.** Typical cases of machine learning models in landslide disaster prediction.

| References | Research Content | Model Used | Method | Conclusion |
|---|---|---|---|---|
| Tehrany, M.S. et al. [145] | Evaluation of flood sensitivity | Support vector machine (SVM) and frequency ratio (FR) | Propose a new integration method, in which the spatial modeling is established in the flood sensitivity evaluation by integrating support vector machine (SVM) and frequency ratio (FR) | The proposed integration method has fast, accurate, and reasonable effectiveness in the flood sensitivity evaluation. |

**Table 8.** *Cont.*

| References | Research Content | Model Used | Method | Conclusion |
|---|---|---|---|---|
| Huang, L. et al. [146] | Weather early warning of precipitation-induced landslides | Deep learning | Propose deep belief network (DBN) method with Softmax classifier and Dropout mechanism, in which the Softmax classifier is added to the top of the DBN neural network to improve the prediction accuracy | Compared with the existing BP neural network algorithm and the BP algorithm based on particle swarm optimizer (PSO-BP) algorithm, the newly proposed method has higher accuracy and better technical performance. |
| Luo, X. et al. [147] | Generation of landslide sensitivity map | Random subspace (RS) and logistic model tree (LMT) | Propose a hybrid machine learning method RSLMT, in which the landslide sensitivity map (LSM) is generated by coupled random subspace (RS) and logistic model tree (LMT) | The uncertainty introduced by the characteristics is input, and the over-fitting problem is solved by dimension reduction to improve the prediction rate of landslide occurrence. |
| Sahin, E.K. et al. [148] | Landslide sensitivity map | Integration method based on regression tree | Draw landslide sensitivity maps by three integration methods based on regression tree such as gradient boosting machine (GBM), extreme gradient boosting (XGBoost), and random forest (RF) | The prediction ability of the model created by the optimal factor combination is the highest. |
| Di Napoli, M. et al. [149] | Statistics of landslide sensitivity map | ML algorithm of artificial neural network, generalized lifting model, and maximum entropy | A new methodology is proposed and tested in the study area, and the eliciting factor is selected for evaluation by a variance inflation factor | Integrated modeling based on artificial neural network, generalized boosting model, and maximum entropy ML algorithm, showing higher reliability. |

In recent years, remote sensing data classification methods have gradually developed in the direction of machine learning, among which deep learning methods for processing remote sensing data and pattern classification in landslide susceptibility mapping have gradually emerged [150]. A deep learning approach is an automatic model building method for analyzing data, making it possible to learn the fundamental relationships and hidden observations present in the data to build an analytical model [151]. Trong-An proposed a system that combines deep learning and image transformation algorithms to detect the location of landslides in satellite images [152]. In order to accurately identify landslides under different lighting conditions, they classified the landslide images by using a transformation algorithm Hue-Bi-dimensional empirical model decomposition (H-BEMD) to determine the area and size of landslides. The results showed that the accuracy of the method in the classification process was as high as 96%. Guang Xu proposed a recognition model of land cover types in field photos based on deep learning technology [153], which has good recognition accuracy for land cover classification.

## 5. Discussion

The landslide disaster system is complex with dynamic development, nonlinearity, and uncertainty. Furthermore, the disaster formation process involves a multi-hierarchy structure, multiple disaster-causing factors, and multiple control parameters. Therefore, when analyzing the uncertainty of landslide disaster prediction, this paper made the following discussions and suggestions in four subfields:

The uncertainty analysis of disaster-causing factors must be made on the basis that the main disaster-causing factors are apparent. Otherwise, the nonlinear correlation analysis between landslide disasters and disaster-causing factors will become a black-box problem,

and the prediction uncertainty analysis will lose its premise. Although there is no unified standard to include main disaster-causing factors into the research scope, the general principle is that when all factors and their correlations cannot be fully considered, it is essential to grasp the controlling effects of the main disaster-causing factors on landslide disaster prediction and ignore the influence of the secondary factors. The selection of the main disaster-causing factors should consider the difficulty of data acquisition, relevance to the research content, similar cases, geological conditions, and environmental factors, and other aspects.

Regarding the uncertainty analysis of prediction units, a good state division of the secondary factors of disaster-causing conditions plays a certain role in optimizing the prediction models, reducing the prediction uncertainty, and improving the prediction effects. When grid division is adopted, it is suggested to rationalize the state of continuous variables according to the principle of difference rather than divide them in an equally-spaced way to optimize the model and avoid the deviation caused by subjective judgment. The division by irregular units is usually very complex, but to reduce the impact of the uncertainty of prediction units on the prediction effects, diversified polygon shapes (e.g., irregular ellipse, dustpan, semicircle, strip, etc.) need to be selected to meet the geological rationality and statistical randomness. In addition, although the vector superposition method based on GIS technology does not need to consider the error caused by boundary differences, there is a big gap between the famous version and the commercial GIS tool. Therefore, integrated secondary development will be the mainstream direction of the GIS system in the uncertainty analysis of prediction units, which can not only improve the efficiency of the application of the GIS system but also be convenient for researchers to exert their imagination and later maintenance.

Regarding the uncertainty of model space data sets, the proportion of the training set and a test set of the data has a particular impact on the modeling accuracy of uncertainty analysis for landslide disaster prediction. However, there is no theoretical research demonstration for determining which proportion to choose, so ratios such as 1:9, 2:8, 3:7, 4:6, and 5:5 have been selected before. When the training set is much less than the test set, it will be difficult to fully reflect the correlation law between the research samples and the uncertainty analysis content of landslide disasters, while when the training set is greater than the test set, it will be difficult for the model accuracy to reflect the accuracy of landslide prediction. Moreover, the proportion of the training set is also affected by the size of the original sample library, but there is little research in this area. Therefore, it is suggested to conduct more tests and repeatedly adjust the proportion when the uncertainty of landslide prediction is analyzed to optimize the ratio of the training set and the test set as much as possible under the condition of ensuring the prediction accuracy.

In terms of the uncertainty analysis of prediction models, even if the prediction accuracy of different prediction models is nearly equal, the distribution characteristics of their susceptibility indexes will be very different. Therefore, the existing research is challenging to give generally accepted conclusions on which model has better performance, which model has more reliable accuracy, and which prediction model is more conducive to landslide disaster modeling. However, through sorting out the documents in this paper, it was found that various coupling models, integration models, and hybrid models have more advantages in model fitting and prediction performance, but these models often make the model design and calculation complicated. Therefore, it is suggested that when multiple models need to be involved in the process of landslide disaster analysis, the model shall be simplified as much as possible while ensuring accuracy and reliability to promote the application of the model and guide the work and practice of landslide disaster prevention and control.

## 6. Conclusions

There are many uncertainties in predicting landslide disaster susceptibility, such as randomness, fuzziness, instability, etc. Based on extensive document analysis, this paper

created a bibliometric analysis from three aspects (statistics of documents publication time, contribution analysis, and analysis of highly cited documents) through comprehensively using bibliometric analysis technology and knowledge mapping software tools, making a series of maps from three aspects (scientific research cooperation, research hotspots, and frontier trend), combined with professional knowledge for knowledge mapping analysis, and drew the following conclusions:

(1) In terms of the number of a document published, the research on the uncertainty of landslide susceptibility prediction shows an increasing trend, which can be divided into the rise stage of research (1982–2005), the apparent growth stage (2006–2016), and the vigorous development stage (2017–2022) (Figure 2).

(2) From the contribution analysis, it was found that Guzzetti F team has the highest number of documents published, and the documents they published are the most authoritative (Table 2); the three most influential countries in this field are China, Italy, and the United States (Figure 3); documents and journals came from a variety of sources, among which ENGINEERING GEOLOGY, LANDSLIDES, and GEOMORPHOLOGY have the largest number of publications (Figure 4).

(3) Through the analysis of research hotspots and development trends, the influencing factors of debris flow, underlying surface structure, and rainfall are the research hotspots, and random forest, machine learning, and neural networks are the frontier research trends in the future.

On this basis, this paper systematically summarized the research progress and development trend in the uncertainty analysis of landslide susceptibility prediction from four key research subfields (such as disaster-causing factors, prediction units, model space data sets, and prediction models), discussed the main problems encountered in current research of several subfields, and put forward some suggestions. In general, the research on the uncertainty of landslide susceptibility prediction is still at an early stage of development. Due to the complexity of the landslide disaster itself, there are still some problems for the future, such as selection of the main disaster-causing factors, data set proportion optimization, and model selection. The landslide susceptibility mapping models based on integrations of metaheuristic optimization and machine learning should be summarized as a research trend. The fuzzy logic method can comprehensively take into account the requirements of various landslide disaster-causing factors, while the intelligent learning method has low requirements for data accuracy but high calculation efficiency; therefore, these two methods will become some mainstream research directions in this field in the future.

**Author Contributions:** Z.Y. and H.L. drafted the manuscript and were responsible for the research design, experiment, and analysis. C.L., W.L., R.N., W.Z., Z.Z., D.Z. and M.Z. reviewed and edited the manuscript. L.Z., G.L., X.D., S.M., X.F. and Z.R. supported the data preparation and the interpretation of the results. All of the authors contributed to editing and reviewing the manuscript. All authors have read and agreed to the published version of the manuscript.

**Funding:** This research was supported by the National Key R&D Program of China (2019YFC1510700), the National Natural Science Foundation of China (41701499), the funding provided by the Alexander von Humboldt-Stiftung, the Sichuan Science and Technology Program (2018GZ0265), the Geomatics Technology and Application Key Laboratory of Qinghai Province, China (QHDX-2018-07), the Major Scientific and Technological Special Program of Sichuan Province, China (2018SZDZX0027), and the Key Research and Development Program of Sichuan Province, China (2018SZ027, 2019-YF09-00081-SN).

**Data Availability Statement:** The data that support the findings of this study are available from the corresponding author upon reasonable request.

**Conflicts of Interest:** The authors declare no conflict of interest.

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
