# Peer review of "Research on Uncertainty of Landslide Susceptibility Prediction—Bibliometrics and Knowledge Graph Analysis"

_remotesensing, doi:10.3390/rs14163879_

Round 1
Reviewer 1 Report
This paper aims to summarize and analyze previous studies to reduce the impact of uncertain factors on the prediction results of landslide susceptibility mapping. To achieve the goal, the authors have reviewed 600 documents collected by the two data platforms of Web of Science and Scopus in the past 40 years. The disaster-causing factors, prediction units, model space data sets, and prediction models are the subjects of interest and have been systematically reviewed. In the conclusion, the authors have pointed out some suggestions to provide references for further improving the prediction accuracy of landslide susceptibility mapping. In general, the authors have done a good review work and the paper has summarized the main trends in landslide susceptibility prediction.
After reading the paper, the reviewer has the following comments:
1) Data sampling/collection processes are crucial for landslide susceptibility mapping. Consider adding one or several paragraphs to discuss/review these issues.
2) The construction of landslide inventories is also very important for landslide susceptibility mapping. It is beneficial to review related papers and provide one or several paragraphs to discuss/review/summarize this issue.
3) Review/discussion regarding the landslide influencing factors used in the previous works can also be added/improved in the manuscript. A graph or a table can be used to summarize the used factors.
4) With regard to machine learning models, as shown in Fig. 16, the determine of their hyper-parameters are crucial. A review of methods for setting the hyper-parameters of models used for landslide susceptibility mapping can be added or elaborated. Herein, the landslide susceptibility mapping models based on integrations of metaheuristic optimization and machine learning should be summarized as a research trend.
5) More works related to the emerging deep learning methods used for processing remote sensing data and pattern classification in landslide susceptibility mapping should be included and reviewed.
Finally, it is a very nice review paper and congratulation on the authors’ work.
Reviewer 2 Report
The paper presents uncertainties overview in predicting landslide disaster susceptibility. The trend in the uncertainty analysis of landslide susceptibility prediction is given. Four aspects are given into account: disaster-causing factors, prediction units, model space data sets, and prediction models.
Good literature overview and analysis.
However, from my point of view, it would be great to add more about the types of uncertainty: fuzzy, noisy data, inconsistency etc. It would be grat to add model to deal with inconsistency too.
Round 2
Reviewer 1 Report
I have no further comments.
Reviewer 2 Report
The authors took into account comments. Thank you.